# CombinationTS: A Modular Framework for Understanding Time-Series Forecasting Models

**Xiaorui Wang** [1 2] **Fanda Fan** [† 1] **Chenxi Wang** [1 2] **Yuxuan Yang** [1 2] **Rui Tang** [3] **Kuoyu Gao** [4] **Simiao Pang** [4] **Yuanfeng Shang** [1] **Zhipeng Liu** [4] **Wanling Gao** [1] **Lei Wang** [1] **Jianfeng Zhan** [1 2]

## Abstract

Recent progress in time-series forecasting has led to rapidly increasing architectural complexity, yet many reported State-of-the-Art gains are statistically fragile or misattributed. We argue that progress requires a shift from model selection to modular attribution, identifying which components truly drive performance. We propose CombinationTS, a self-contained probabilistic evaluation framework grounded in Evaluatology's view of evaluation as effect attribution. It decomposes forecasting models into orthogonal modules—Input Transformation, Embedding, Encoder, Decoder, and Output Transformation—and evaluates them under a shared evaluation condition space. By quantifying each component via marginalized performance ($\mu$) and stability ($\sigma$), CombinationTS enables robust attribution beyond fragile point estimates. Through large-scale paired evaluation, we uncover the Identity Paradox: once the data view (Embedding) is well-designed, a parameter-free Identity Encoder often matches or outperforms complex backbones. We further show that explicit structural priors introduced via Input Transformations yield a more favorable performance–stability trade-off than increasing Encoder complexity, establishing a principled baseline for architectural necessity. The code is available at https://github.com/BenchCouncil/CombinationTS.

---

[1]Institute of Computing Technology, Chinese Academy of Sciences, Beijing, China [2]University of Chinese Academy of Sciences, Beijing, China [3]Beijing Normal University - Hong Kong Baptist University United International College, Zhuhai, China [4]Northeastern University, Shenyang, China. Correspondence to: Fanda Fan <fanfanda@ict.ac.cn>.

*Proceedings of the 43rd International Conference on Machine Learning*, Seoul, South Korea. PMLR 306, 2026. Copyright 2026 by the author(s).

## 1. Introduction

Time-series forecasting drives high-stakes decisions in finance (Sezer et al., 2020; Liu et al., 2025a;c), energy (Pinto et al., 2021), healthcare (Kaushik et al., 2020), transportation (Sun et al., 2023), and climate science (Du et al., 2019). The landscape of Time-Series Forecasting (TSF) has undergone rapid architectural expansion, with models becoming increasingly complex in both structure and design. Importantly, this complexity has not arisen solely from deeper or more powerful modeling modules, but from a progressive shift toward richer *data view* (Embedding) and *Input Transformation*.

Recent advances move from sparse-attention Transformers (Zhou et al., 2021; 2022) to patch-based tokenization (Nie et al., 2023; Stitsyuk & Choi, 2025; Hu et al., 2025), channel-wise embeddings (Liu et al., 2024; Wang et al., 2024b; Qiu et al., 2025c), multi-scale MLP-based designs (Wang et al., 2024a; Tang & Zhang, 2025), and attention-modulation approaches (Liu et al., 2026c), all of which substantially reshape how data is presented to the model rather than how it is reasoned over. Such a trend of escalating architectural complexity is not confined to forecasting alone; analogous patterns of multi-scale redundancy and domain robustness challenges have been observed across broader time series analysis tasks, including classification (Liu et al., 2025b; 2026b).

While these developments report steady gains on public leaderboards, they also entangle architectural design with data preprocessing choices and evaluation configurations, making performance improvements increasingly difficult to attribute. Recent studies show that the reported gap between State-of-the-Art models and strong baselines is often comparable to variability induced by random seeds or hyperparameter alignment (Tan et al., 2024; Brigato et al., 2026), raising a fundamental question: *Do modern TSF gains reflect genuine architectural reasoning, or are they largely driven by improved data view and favorable evaluation conditions?*

This ambiguity stems from two fundamental methodological deficits in current benchmarking practices:

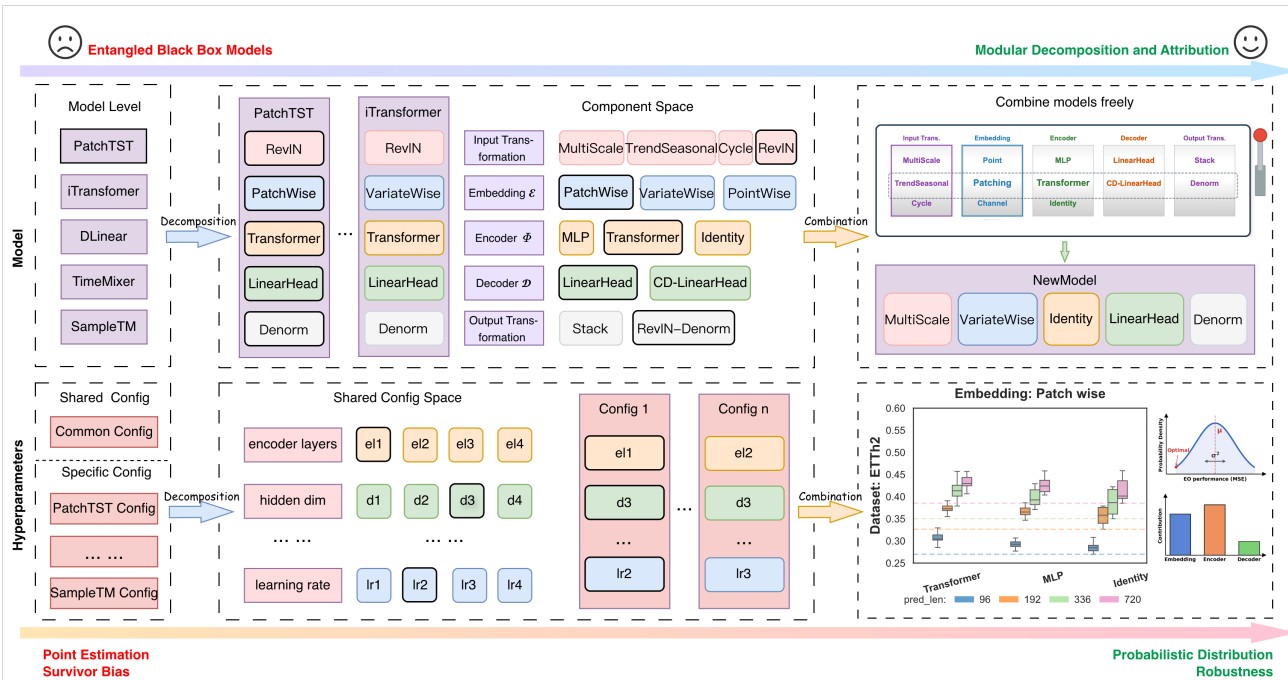

*Figure 1.* The conceptual framework of CombinationTS. The diagram illustrates the paradigm shift proposed in this work across two dimensions. (Top) Model Architecture: We transition from entangled Black Box models to a modular decomposition framework. By dismantling models into five orthogonal components—Input Transformation, Embedding, Encoder, Decoder, and Output Transformation—we enable free recombination to identify the true sources of performance attribution. (Bottom) Evaluation Paradigm: We shift from point estimation, which is susceptible to survivor bias, to probabilistic distribution–based evaluation. Instead of fixing model-specific configurations or a shared configuration, we assess the robustness of the evaluated object over a shared configuration space, using distributional metrics to ensure fair and comparable evaluation.

(i). The Attribution Gap (Monolithic vs. Modular): Existing evaluations treat models as indivisible "black boxes" (Shao et al., 2024; Qiu et al., 2024; 2025a; Wang et al., 2026), conflating the contribution of the *data view* (Embedding) with that of the *Encoder* (e.g., Attention, MLP). Consequently, it remains unclear whether a model's success is driven by *how it sees data* or *how it processes it*.

(ii). The Benchmarking Crisis (Point vs. Distribution): Standard protocols (Wu et al., 2023; Wang et al., 2024a) often fall into the "Fairness Trap" (using suboptimal fixed settings) or the "Best Trap" (cherry-picking peak results). The "Best Trap" arises because models with more tunable hyperparameters have a larger search space, making them more likely to stumble upon a favorable configuration and report a deceptively strong peak that does not reflect expected performance. Conversely, the "Fairness Trap" arises because constraining all models to the same fixed configuration may suppress the effective operating regime of architectures that require specific hyperparameter settings, systematically underestimating their potential. Either way, such point estimates fail to capture the *expected performance* ($\mu$) and *stability* ($\sigma$) of a design under the epistemic uncertainty of real-world hyperparameters.

To resolve these deficits, we advocate a shift from *Model Selection* to *Modular Attribution*, and construct a self-contained evaluation system for time-series forecasting. Rather than proposing yet another architecture, **CombinationTS** reformulates model analysis as a component-level attribution problem. Specifically, we decompose the forecasting pipeline into five orthogonal stages—*Input Transformation, Embedding, Encoder, Decoder, and Output Transformation*—so that architectural effects can be isolated and recombined in a controlled manner. Grounded in the principles of *Evaluatology* (Zhan et al., 2024; 2025), we characterize evaluation as a stochastic process by defining a shared Evaluation Condition Space $\Omega$, which captures variations in complementary modules and training configurations. By performing stratified Monte Carlo sampling over $\Omega$, we estimate the expected effectiveness and stability of each component, enabling robust attribution that is resilient to configuration noise.

Through this rigorous probabilistic audit, we present three empirical findings that challenge prevailing intuitions:

❶ The "Identity" Paradox (data view (Embedding) > modeling module (Encoder)): Within the explored condition space $\Omega$, we observe that once the *data view* (Embedding)

is well-designed, a parameter-free **Identity Encoder** frequently achieves comparable or even superior effectiveness and stability, outperforming complex Transformers and Mixers. This suggests that for many standard benchmarks, the primary performance bottleneck lies in data view formulation rather than modeling module.

❷ Conditional Utility of Structural Priors (Input Transformation): We find that Input Transformations are not universally superior. While explicit *periodic inductive biases* (e.g., learnable cycles (Lin et al., 2024)) provide consistent gains, decomposition strategies (e.g., multi-scale splitting) only outperform the RevIN baseline when coupled with sophisticated *cross-component interaction*. Naive decomposition often degrades performance, highlighting the necessity of architectural synergy.

❸ Effectiveness Gain of Spectral Modeling: Our results reveal that frequency-domain modeling (e.g., Fourier/Wavelet) yields higher effectiveness ($\mu$) than temporal baselines, yet provides no significant reduction in instability ($\sigma$), suggesting that the spectral advantage stems from better signal representation rather than robustness to hyperparameter variation.

Our work offers a standardized protocol for architectural auditing: (i). **Framework**: We propose CombinationTS, a modular framework that decouples the TSF pipeline into five interchangeable stages, enabling the combinatorial analysis of $100+$ architectural variants. (ii). **Protocol**: We establish a probabilistic evaluation protocol based on EC-sampling, shifting the metric from fragile point estimates (MSE) to robust distributional statistics ($\mu, \sigma$). (iii). **Systematic Evaluation**: We provide a systematic evaluation of modern components, identifying the "Identity Paradox" and establishing a new burden of proof for future architectural complexity. (iv). **Open Source**: We release the modular library and the full log of experimental configurations to facilitate reproducible component-level research.

## 2. Related Work

**Evolution of TSF Architectures.** The landscape of Time-Series Forecasting (TSF) has shifted from rigorous architectural complexity to a focus on data-centric mechanisms. Early Transformer-based models, such as **Informer** (Zhou et al., 2021) and **Autoformer** (Wu et al., 2021), focused on optimizing the Encoder efficiency via sparse attention or decomposition. However, the emergence of **DLinear** (Zeng et al., 2023) challenged this direction, demonstrating that simple linear layers combined with trend-seasonal decomposition could achieve state-of-the-art performance. Subsequently, the focus shifted to the *data view* (Embedding): **PatchTST** (Nie et al., 2023) introduced patch-wise tokenization to capture local semantics, while **iTransformer** (Liu et al., 2024) proposed an inverted variate-wise embedding

to explicitly model multivariate correlations; **DUET** (Qiu et al., 2025c) further enhances this through dual clustering, and attention modulation (Liu et al., 2026c) improves Transformer expressiveness. Most recently, architectures like **TimeMixer** (Wang et al., 2024a) and **CycleNet** (Lin et al., 2024) further emphasize the role of Input Transformations—incorporating multi-scale downsampling and explicit periodicity—over deep Encoder design. Complementarily, decomposition-guided objectives (Qiu et al., 2025b) and simplicity-first baselines (Liu et al., 2026a; Zeng et al., 2023) challenge the necessity of architectural complexity.

In this work, rather than proposing a new architecture, we select these representative models not as monolithic competitors, but as exemplars of distinct functional modules—Input Transformation, Embedding, Encoder, Decoder and Output Transformation—to rigorously dissect the true source of their effectiveness.

**Benchmarking and Evaluation Paradigms.** The "reproducibility crisis" in machine learning has prompted a critical re-examination of how forecasting models are evaluated. While libraries like **TSLib** (Wang et al., 2026), **BasicTS** (Shao et al., 2024), **TFB** (Qiu et al., 2024), and **TAB** (Qiu et al., 2025a) have standardized implementation interfaces for time series forecasting, recent studies highlight that code unification alone is insufficient to guarantee rigorous comparison. Chen et al. (2025b) scrutinize the inner workings of Transformers, revealing that performance gains are frequently misattributed to complex reasoning mechanisms when they actually stem from simple *data view* choices (e.g., intra-variate modeling). Furthermore, Brigato et al. (2026) explicitly argue that *"There are no Champions"* in long-term forecasting. Through an exhaustive audit of over 3,500 models, they demonstrate that "SOTA" claims are statistically fragile, often flipping with minor changes in evaluation contexts. Similarly, spectral and frequency-domain methods (Chen et al., 2025a; Stitsyuk & Choi, 2025) have recently gained traction, yet their necessity relative to simpler alternatives remains under-examined. These findings underscore a fundamental flaw in the current "Leaderboard" paradigm: ranking models based on point estimates (e.g., a single random seed) is scientifically untenable.

Addressing this, our work adopts the principles of *Evaluatology* (Zhan et al., 2024; 2025) to shift the focus from ranking models to **auditing mechanisms**. While existing benchmarking tools compare models as monolithic entities, none provides the ability to *attribute* performance differences to specific architectural components under controlled, probabilistic evaluation—a gap that CombinationTS directly addresses. Instead of relying on fragile point comparisons, CombinationTS employs a Probabilistic Evaluation Protocol. By treating hyperparameters and experimental settings as stochastic Evaluation Conditions (EC) sampled from a

broad distribution, we quantify the *marginalized performance* ($\mu$) and *stability* ($\sigma$) of specific components, establishing a statistically defensible framework to rigorously measure the effectiveness-stability trade-off.

# 3. The CombinationTS Framework

To transition from "Model Selection" to "Modular Understanding," we must simultaneously address two fundamental methodological deficits in current research:

**The Attribution Gap.** The monolithic treatment of forecasting models obscures the source of performance gains, making it difficult to determine whether success stems from a novel encoder or simply a superior data view.

**The Benchmarking Crisis.** The reliance on point estimates leads to two fundamental traps, yielding conclusions that are statistically fragile and often unreproducible. First, the *"Fairness" Trap*: constraining all models to a fixed setting may fail to activate the effective operating regime of specific architectures, systematically underestimating their potential. Second, the *"Best" Trap*: reporting the best observed result is statistically fragile, as peak performance may arise from incidental hyperparameter alignment or randomness, thereby misrepresenting expected performance in practice.

Our framework, **CombinationTS**, mitigates these issues by intersecting **Modular Decomposition** (addressing the attribution gap) with a **Probabilistic Evaluation Protocol** (addressing the benchmarking crisis).

## 3.1. Modular Decomposition for Structural Attribution

We consider the multivariate time-series forecasting problem, mapping historical observations $\mathbf{X} \in \mathbb{R}^{T \times N}$ to future predictions $\mathbf{Y} \in \mathbb{R}^{P \times N}$ via a parameterized function $f$. To systematically dissect the internal mechanisms of $f$ rather than treating it as a monolithic "black box," we formalize the forecasting pipeline as a composite of five stages:

$$f = \mathcal{T}_{out}^{-1} \circ \mathcal{D} \circ \Phi \circ \mathcal{E} \circ \mathcal{T}_{in}. \tag{1}$$

Consequently, the forward pass is expressed as

$$\hat{\mathbf{Y}} = f(\mathbf{X}) = (\mathcal{T}_{out}^{-1} \circ \mathcal{D} \circ \Phi \circ \mathcal{E} \circ \mathcal{T}_{in})(\mathbf{X}). \tag{2}$$

This formulation allows us to isolate specific components for analysis:

**Input Transformation ($\mathcal{T}_{in}$).** Injects structural priors into the raw signal (e.g., normalization, detrending, or trend-seasonal decomposition). *Operationally*, $\mathcal{T}_{in}$ is restricted to transformations applied directly on $\mathbf{X}$ before tokenization and admits an inverse $\mathcal{T}_{out}^{-1}$ when applicable.

**Embedding ($\mathcal{E}$).** Defines the **data view** (tokenization strategy). To unify diverse views (e.g., variate tokens vs. tem-

poral patches), we standardize the latent representation into a unified tensor interface $\mathcal{Z} = \mathcal{E}(\mathcal{T}_{in}(\mathbf{X})) \in \mathbb{R}^{B \times C \times L \times D}$, where $B$ is the batch size, $C$ denotes the **spatial dimension** (number of variates/channels), $L$ denotes the **temporal dimension** (number of temporal tokens), and $D$ is the hidden channel size. *Crucially*, $\mathcal{E}$ only determines the view and local within-token encoding, while any cross-token dependency modeling is delegated to the encoder $\Phi$.

**Encoder ($\Phi$).** The modeling module operating on $\mathcal{Z}$ (e.g., self-attention, MLP-mixer), responsible for capturing interactions among tokens and/or within-token contexts.

**Decoder ($\mathcal{D}$).** Projects the processed latent features to the forecasting horizon, e.g., mapping encoder outputs to $\mathbb{R}^{P \times N}$ (or an intermediate horizon-aligned representation).

**Output Transformation ($\mathcal{T}_{out}^{-1}$).** The inverse operation of $\mathcal{T}_{in}$ (when applicable), e.g., adding back a removed trend component or applying inverse normalization to recover the original scale.

## 3.2. Probabilistic Evaluation for Robust Benchmarking

Standard benchmarking practices in time-series forecasting typically fix a single configuration to report a single-value performance metric. Such point-estimate–based evaluation suffers from the "Fairness Trap" and the "Best Trap", both failing to capture the expected performance and stability of a design.

To overcome these limitations, we build a self-contained evaluation system that treats architectural analysis as a signal–noise disentanglement problem: the *signal* is the component whose effect we aim to attribute, while the *noise* arises from evaluation conditions. Grounded in Evaluatology (Zhan et al., 2024; Wang et al., 2025; Zhan et al., 2025), this view naturally leads to a probabilistic protocol—we move beyond point estimates and instead compare performance distributions induced by evaluation environments under a shared condition space.

Concretely, we define the evaluation system through two complementary entities:

**Evaluated Object.** [1] The deterministic signal we aim to attribute, corresponding to the module variant under investigation at a specific stage of the forecasting pipeline (e.g., $\theta = $ *Variate-wise Embedding*).

**Evaluation Condition.** [2] The stochastic environment under which an EO operates. It encapsulates *all* factors that

---

[1]**EO**: We denote the evaluated object, i.e., the architectural component whose contribution is being analyzed, by $\theta$.

[2]**EC**: We denote the evaluation condition by $\mathbf{c}$.

may influence the observed performance once $\theta$ is fixed, including (i) complementary module instantiations in the remaining stages, (ii) training hyperparameters, and (iii) sources of randomness. Concretely, $\mathbf{c}$ includes (but is not limited to) random seeds, batch size, dropout rate, learning rate schedules, initialization, and other training or evaluation choices. We treat $\mathbf{c}$ as a random variable sampled from a broad evaluation condition space $\Omega$, rather than as a fixed configuration or a quantity to be optimized away.

**Statistical Attribution Target.** Under this formulation, the performance of a module $\theta$ is modeled as a random variable $L(\theta, \mathbf{c})$ governed by $\mathbf{c} \sim \Omega$. We quantify the contribution of $\theta$ using two distributional statistics:

$$\mu(\theta) = \mathbb{E}_{\mathbf{c}\sim\Omega}[L(\theta, \mathbf{c})], \quad \sigma(\theta) = \sqrt{\mathrm{Var}_{\mathbf{c}\sim\Omega}[L(\theta, \mathbf{c})]}, \tag{3}$$

where $\mu$ denotes **Marginalized Performance**, capturing expected performance across evaluation conditions, and $\sigma$ denotes **Stability**, measuring sensitivity to such conditions. A superior module is therefore expected to improve $\mu$ without incurring a disproportionate increase in $\sigma$.

### 3.3. Stratified Monte Carlo Estimation

To obtain robust and comparable estimates, we adopt a **stratified** and **paired** Monte Carlo protocol. We construct a fixed evaluation condition set $\{\mathbf{c}_k\}_{k=1}^{K}$ sampled from $\Omega$, stratified across datasets and prediction horizons. Crucially, to eliminate confounding factors arising from stochastic sampling, we employ a **paired design**: every Evaluated Object (EO) is assessed under the *exact same* set of sampled conditions.

Based on these $K$ shared samples, we report three key statistics for each module $\theta$:

**Marginalized Performance ($\hat{\mu}$).** The sample mean of the loss over the sampled conditions:

$$\hat{\mu}(\theta) = \frac{1}{K}\sum_{k=1}^{K} L(\theta, \mathbf{c}_k). \tag{4}$$

**Stability ($\hat{\sigma}$).** The sample standard deviation of the loss over the sampled conditions:

$$\hat{\sigma}(\theta) = \sqrt{\frac{1}{K-1}\sum_{k=1}^{K}\left(L(\theta, \mathbf{c}_k) - \hat{\mu}(\theta)\right)^2}. \tag{5}$$

**Peak Potential ($L_{best}$).** To explicitly contrast with the "Best Trap", we also report the single best performance observed during sampling:

$$L_{best}(\theta) = \min_{k} L(\theta, \mathbf{c}_k). \tag{6}$$

By reporting both the expected ($\hat{\mu}$) and peak ($L_{best}$) performance, we can rigorously determine whether a model's

SOTA claim is a generalized capability or a rare outlier.

## 4. Experiments

In this section, we transition from theoretical formulation to empirical attribution. Applying the **CombinationTS** framework (illustrated in Figure 1), we systematically deconstruct the "black box" of modern time-series forecasting models to isolate the true drivers of performance. Our experimental suite is built upon a **highly customized version** of the Time Series Library (TSLib) (Wang et al., 2026), extensively modified to enable the **combinatorial assembly** of decoupled modules—serving as the implementation engine for **CombinationTS**. We conduct extensive evaluations across **six** widely adopted benchmarks: **Weather**, **Electricity**, and the four subsets of the **ETT** dataset (ETTh1, ETTh2, ETTm1, ETTm2).

Rather than chasing State-of-the-Art (SOTA) on a fixed leaderboard, our experiments are designed to answer three fundamental questions that directly address the **Attribution Gap** and **Benchmarking Crisis** identified in the Introduction:

**RQ1: Dissecting the Backbone (Exp. 1).** Which drives forecasting performance—data view (Embedding $\mathcal{E}$) or modeling module (Encoder $\Phi$)? We evaluate all pairwise ($\mathcal{E}, \Phi$) combinations under a unified probabilistic protocol, measuring $\hat{\mu}$, $\hat{\sigma}$, and $L_{best}$.

**RQ2: Auditing Input Transformations (Exp. 2).** Can Input Transformations ($\mathcal{T}_{in}$)—decomposition, downsampling, or cyclic embeddings—substitute for complex encoders? We compare four representative strategies across an expanded EC space ($D \in \{16, \ldots, 512\}$) to test whether Input Transformations enable efficient forecasting without over-parameterization.

**RQ3: Revisiting Spectral Processing (Exp. 3).** Does frequency-domain modeling offer performance advantages over time-domain processing, and does it also improve stability? We conduct a controlled pair (SimpleTM vs. iTransformer) under matched Variate-wise Embedding to determine whether spectral processing provides genuine gains in $\mu$ or $\sigma$ beyond what the Identity baseline already achieves.

**Evaluation Protocol.** We adopt MSE as the primary metric $L$ and report three statistics: **Marginalized Performance ($\hat{\mu}$)**, **Stability ($\hat{\sigma}$)**, and **Peak Potential ($L_{best}$)**. To isolate configurational robustness from initialization noise, the random seed is fixed across all runs and dropout is set to 0.1. Multi-seed verification confirms that this choice does not affect our conclusions (Section A.7). Training is standardized at batch size 32, 30 epochs, and early-stopping patience 3, ensuring any observed variance is attributable solely to architectural and hyperparameter choices.

## 4.1. Dissecting the Backbone: The Hierarchy of View and Representation

Building upon the modular decomposition in Eq. 1, we focus on the architectural "backbone" of modern forecasting models: the interplay between **data view (Embedding $\mathcal{E}$)** and **modeling module (Encoder $\Phi$)**.

**Controlled Factors.** To isolate the backbone effect, we fix the Input and Output Transformations ($\mathcal{T}_{in}, \mathcal{T}_{out}^{-1}$) to **RevIN** (Kim et al., 2021) for all models, and systematically evaluate combinations of **Embedding** and **Encoder**, together with a lightweight **Decoder**. Importantly, **each encoder is evaluated on *all* embedding strategies** under the same probabilistic protocol (paired EC sampling), avoiding selective reporting.

### 4.1.1. EXPERIMENT SETUP

**Evaluated Objects.** We construct a modular search space for backbone analysis:

$$\Theta_{\text{backbone}} = \mathcal{E} \times \Phi \times \mathcal{D}.$$

A specific EO is a concrete module combination $\theta \in \Theta_{\text{backbone}}$. Table 1 summarizes all EO variants and their corresponding tensor transformations (*omitting batch dimension for simplicity*; the implementation follows a standard batch-first layout).

**Evaluation Conditions.** To estimate $\mu(\theta)$ and $\sigma(\theta)$, we sample EC from a condition space $\Omega$. Each condition $c \in \Omega$ specifies: dataset $d \in \{$Weather, Electricity, ETTh1, ETTh2, ETTm1, ETTm2$\}$, look-back $T \in \{96, 192, 336, 512\}$, horizon $P \in \{96, 192, 336, 720\}$, encoder layers $el \in \{1, 2, 3\}$, latent dim $D \in \{64, 128, 256, 512\}$, and learning rate $\eta \in \{$1e-3, 1e-4$\}$; RevIN (Kim et al., 2021) and a fixed seed are applied uniformly. We construct a fixed set $\Omega_{\text{eval}} = \{c_k\}_{k=1}^{K}$ with $K = 600$ stratified ECs (100 per dataset); all EOs share the same paired EC set for fair comparison, with inactive dimensions ignored.

### 4.1.2. RESULTS AND ANALYSIS

**Embedding (data view) Analysis.** Based on Table 2 and Figure 3, we summarize three insights about **data view** (95% confidence intervals reported in Table 14):

❶ **Robustness of structured tokenization.** Patch-wise and Variate-wise embeddings consistently achieve strong marginalized performance ($\hat{\mu}$) with comparatively low instability ($\hat{\sigma}$) across most datasets and horizons. Explicit inductive bias in $\mathcal{E}$—either local temporal aggregation (Patch) or tokenizing by variates—stabilizes optimization under hyperparameter variation.

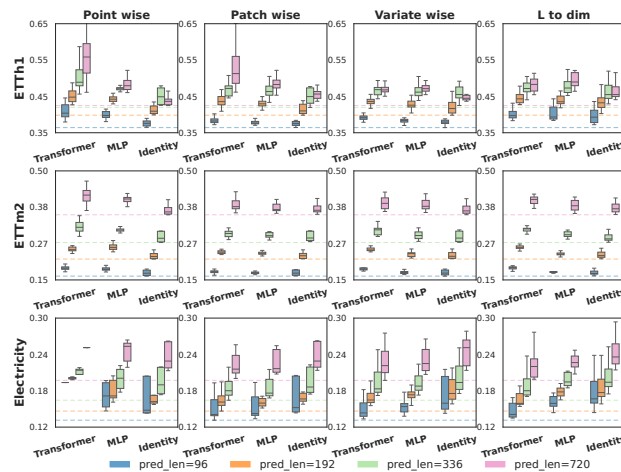

*Figure 2.* Distribution of encoder effectiveness under paired EC sampling (boxplot). The full figure is presented in Figure 4.

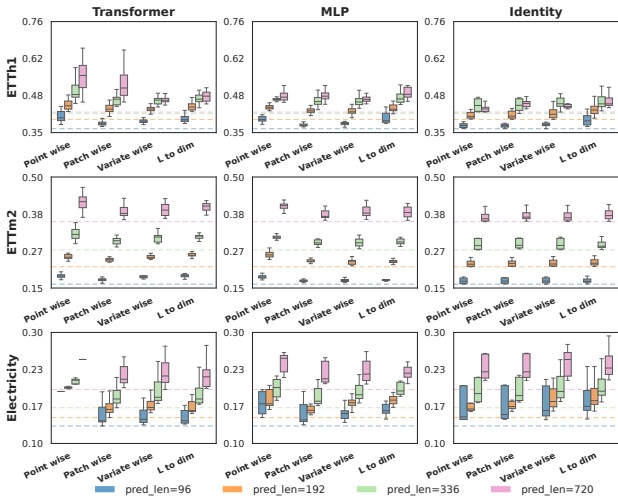

*Figure 3.* Distribution of embedding effectiveness under paired EC sampling (boxplot). The full figure is presented in Figure 5.

❷ **The "std trap" of Point-wise embedding.** Point-wise embedding tends to exhibit worse average performance and higher standard deviation on several ETT subsets, yet can occasionally reach a very low $L_{best}$. This illustrates a key pitfall of "best-result" benchmarking: a volatile design may sporadically hit a favorable configuration despite being unreliable in expectation.

❸ **Efficacy of parameter-free reshaping.** The Time-as-Feature (T→D reshape) heuristic (parameter-free) performs on par with more complex learnable views in many settings. This suggests that the *structure of the tensor* can matter as much as—and sometimes more than—learnable projections of values.

**Encoder (modeling module) Analysis.** Shifting focus to the **Encoder** (Reasoning), Table 3 and Figure 2 reveal the

*Table 1.* **Search Space for Experiment 1 (Backbone Analysis).** We deconstruct the forecasting model into three modular stages. The **Tensor Transformation** column denotes the shape change from Input to Output (assuming input $\mathbf{X} \in \mathbb{R}^{B \times N \times T \times 1}$, output $\mathbf{Y} \in \mathbb{R}^{B \times N \times P \times 1}$), where $B$: batch size, $N$: variates, $T$: look-back length, $P$: prediction length, $S$: patch embedding stride, and $D$: latent dim. $N'$, $T'$, $D'$ depend on the embedding choice. For brevity, the $B$ dimension is omitted in the table.

| Module (EO) | Variant | Mechanism Description | Tensor Transformation |
|---|---|---|---|
| **Embedding** ($\mathcal{E}$) | Point-wise | Projects each time step independently | $\mathbb{R}^{N \times T \times 1} \rightarrow \mathbb{R}^{N \times T \times D}$ |
| | Patch-wise | Aggregates local segments into tokens | $\mathbb{R}^{N \times T \times 1} \rightarrow \mathbb{R}^{N \times \lceil T/S \rceil \times D}$ |
| | Variate-wise | Treats each variate history as one token | $\mathbb{R}^{N \times T \times 1} \rightarrow \mathbb{R}^{N \times 1 \times D}$ |
| | Identity | Preserves raw inputs (No projection) | $\mathbb{R}^{N \times T \times 1} \rightarrow \mathbb{R}^{N \times T \times 1}$ |
| | Time-as-Feature ($T \rightarrow D$ reshape) | Reshapes temporal/variate dim to feature (**parameter-free**) | $\mathbb{R}^{N \times T \times 1} \rightarrow \mathbb{R}^{N \times 1 \times T}$ (or equivalent view) |
| **Encoder** ($\Phi$) | Transformer | Self-Attention | $\mathbb{R}^{N' \times T' \times D'} \rightarrow \mathbb{R}^{N' \times T' \times D'}$ |
| | MLP | Global mixing across dimensions | |
| | Identity | Pass-through baseline | |
| **Decoder** ($\mathcal{D}$) | Shared Linear | Shared weights across variates | $\mathbb{R}^{N' \times T' \times D'} \rightarrow \mathbb{R}^{N \times P \times 1}$ |

*Table 2.* Performance comparison on embeddings. For each metric ($\hat{\mu}$ and $L_{best}$): **blue** = best, **blue** = second best; **red** = best, **red** = second best. Detailed results are reported in Table 8.

| Dataset | H | Point wise | | | Patch wise | | | Variate wise | | | L to dim | | |
|---|---|---|---|---|---|---|---|---|---|---|---|---|---|
| | | $\hat{\mu}$ | $\hat{\sigma}$ | $min$ | $\hat{\mu}$ | $\hat{\sigma}$ | $min$ | $\hat{\mu}$ | $\hat{\sigma}$ | $min$ | $\hat{\mu}$ | $\hat{\sigma}$ | $min$ |
| ETTh1 | 96 | 0.3898 | 0.0143 | 0.3651 | 0.3772 | 0.0058 | 0.3645 | 0.3834 | 0.0080 | 0.3641 | 0.3937 | 0.0156 | 0.3721 |
| | 192 | 0.4324 | 0.0166 | 0.3992 | 0.4240 | 0.0133 | 0.3984 | 0.4247 | 0.0128 | 0.3983 | 0.4343 | 0.0161 | 0.4028 |
| | 336 | 0.4651 | 0.0193 | 0.4255 | 0.4575 | 0.0170 | 0.4196 | 0.4589 | 0.0148 | 0.4249 | 0.4693 | 0.0160 | 0.4294 |
| | 720 | 0.4772 | 0.0415 | 0.4248 | 0.4812 | 0.0326 | 0.4366 | 0.4618 | 0.0140 | 0.4366 | 0.4751 | 0.0223 | 0.4409 |
| | avg | 0.4367 | 0.0364 | - | 0.4277 | 0.0370 | - | 0.4285 | 0.0327 | - | 0.4398 | 0.0342 | - |
| ETTm2 | 96 | 0.1800 | 0.0089 | 0.1625 | 0.1728 | 0.0051 | 0.1631 | 0.1758 | 0.0072 | 0.1622 | 0.1779 | 0.0073 | 0.1633 |
| | 192 | 0.2414 | 0.0136 | 0.2170 | 0.2331 | 0.0079 | 0.2179 | 0.2343 | 0.0106 | 0.2184 | 0.2370 | 0.0115 | 0.2180 |
| | 336 | 0.3019 | 0.0172 | 0.2701 | 0.2911 | 0.0105 | 0.2732 | 0.2928 | 0.0135 | 0.2707 | 0.2968 | 0.0134 | 0.2711 |
| | 720 | 0.3981 | 0.0216 | 0.3591 | 0.3813 | 0.0140 | 0.3608 | 0.3863 | 0.0170 | 0.3594 | 0.3892 | 0.0174 | 0.3603 |
| | avg | 0.2663 | 0.0762 | - | 0.2562 | 0.0725 | - | 0.2596 | 0.0723 | - | 0.2612 | 0.0728 | - |
| Electricity | 96 | 0.1609 | 0.0228 | 0.1431 | 0.1512 | 0.0176 | 0.1315 | 0.1527 | 0.0135 | 0.1335 | 0.1573 | 0.0190 | 0.1349 |
| | 192 | 0.1716 | 0.0154 | 0.1574 | 0.1641 | 0.0111 | 0.1465 | 0.1710 | 0.0111 | 0.1555 | 0.1710 | 0.0115 | 0.1542 |
| | 336 | 0.1917 | 0.0170 | 0.1735 | 0.1828 | 0.0137 | 0.1644 | 0.1906 | 0.0153 | 0.1722 | 0.1928 | 0.0170 | 0.1706 |
| | 720 | 0.2342 | 0.0194 | 0.2132 | 0.2236 | 0.0160 | 0.2004 | 0.2293 | 0.0205 | 0.1972 | 0.2218 | 0.0141 | 0.1991 |
| | avg | 0.1855 | 0.0252 | - | 0.1773 | 0.0246 | - | 0.1827 | 0.0249 | - | 0.1851 | 0.0264 | - |

*Table 3.* Performance of 6 datasets across 4 horizons on 3 encoders (paired EC evaluation). Highlighting follows the convention in Table 2. Detailed results are reported in Table 9.

| Dataset | H | MLP | | | Transformer | | | Identity | | |
|---|---|---|---|---|---|---|---|---|---|---|
| | | $\hat{\mu}$ | $\hat{\sigma}$ | $min$ | $\hat{\mu}$ | $\hat{\sigma}$ | $min$ | $\hat{\mu}$ | $\hat{\sigma}$ | $min$ |
| ETTh1 | 96 | 0.4371 | 0.0816 | 0.3700 | 0.4518 | 0.0930 | 0.3721 | 0.3907 | 0.0231 | 0.3641 |
| | 192 | 0.4870 | 0.0827 | 0.4043 | 0.5044 | 0.0988 | 0.4091 | 0.4334 | 0.0306 | 0.3983 |
| | 336 | 0.5186 | 0.0787 | 0.4335 | 0.5239 | 0.0795 | 0.4406 | 0.4684 | 0.0268 | 0.4196 |
| | 720 | 0.5538 | 0.1039 | 0.4533 | 0.5679 | 0.1014 | 0.4504 | 0.4722 | 0.0406 | 0.4248 |
| | avg | 0.4960 | 0.0938 | - | 0.5095 | 0.1004 | - | 0.4392 | 0.0424 | - |
| ETTm2 | 96 | 0.1893 | 0.0197 | 0.1667 | 0.2027 | 0.0265 | 0.1652 | 0.1755 | 0.0087 | 0.1622 |
| | 192 | 0.2525 | 0.0213 | 0.2201 | 0.2610 | 0.0221 | 0.2313 | 0.2341 | 0.0133 | 0.2170 |
| | 336 | 0.3101 | 0.0198 | 0.2738 | 0.3202 | 0.0199 | 0.2794 | 0.2956 | 0.0194 | 0.2701 |
| | 720 | 0.4057 | 0.0232 | 0.3634 | 0.4163 | 0.0265 | 0.3653 | 0.3919 | 0.0270 | 0.3591 |
| | avg | 0.2784 | 0.0759 | - | 0.2909 | 0.0772 | - | 0.2621 | 0.0756 | - |
| Electricity | 96 | 0.1599 | 0.0171 | 0.1338 | 0.1535 | 0.0194 | 0.1315 | 0.1679 | 0.0254 | 0.1428 |
| | 192 | 0.1729 | 0.0130 | 0.1515 | 0.1671 | 0.0129 | 0.1465 | 0.1742 | 0.0161 | 0.1574 |
| | 336 | 0.1925 | 0.0158 | 0.1681 | 0.1858 | 0.0153 | 0.1644 | 0.1939 | 0.0179 | 0.1735 |
| | 720 | 0.2282 | 0.0171 | 0.2070 | 0.2196 | 0.0171 | 0.1972 | 0.2326 | 0.0191 | 0.2132 |
| | avg | 0.1851 | 0.0253 | - | 0.1776 | 0.0242 | - | 0.1888 | 0.0275 | - |

following (95% confidence intervals reported in Table 15):

❶ **The "Identity" Paradox.** Across ETT datasets, **Transformer-based encoders fail to deliver any statistically significant gain over the parameter-free Identity baseline**, incurring substantially higher instability ($\hat{\sigma}$) without improving marginalized performance ($\hat{\mu}$). On standard benchmarks, widely adopted deep modeling modules act primarily as **over-parameterized noise generators** rather than essential feature extractors.

❷ **Transformer's dependence on the data view.** Transformers exhibit pronounced sensitivity to the embedding stage: substantially more stable under structured tokenization (Patch/Variate-wise), but volatile under unstructured views. Self-attention is not a universal feature extractor; it benefits from semantically meaningful tokens.

❸ **MLP Robustness.** MLP encoders demonstrate **consistent stability** comparable to the Identity baseline, offering a reliable middle ground with learnable capacity but without the optimization volatility of attention mechanisms.

**Robustness of the Identity Paradox.** We verify that the Identity Paradox is not an artifact of statistical noise or

limited evaluation scope through three complementary analyses.

**(i) Statistical significance.** To rigorously validate the Identity Paradox, we perform one-tailed Mann–Whitney U tests ($\alpha=0.05$) on the paired EC samples to assess whether the Identity encoder achieves significantly lower MSE than alternative encoders. As shown in Table 4, Identity achieves significantly lower MSE than both Transformer ($p<0.0001$) and MLP ($p=0.0115$) across all datasets combined. At the per-dataset level (Table 18), Identity is significantly better than Transformer in 3/6 datasets (ETTh1, ETTh2, ETTm2) and MLP in 2/6 datasets (ETTh1, ETTh2), with the non-significant datasets (ECL, ETTm1, Weather) showing Identity at worst on par—never significantly worse. This confirms that the Identity Paradox is not an artifact of selective reporting but a statistically defensible finding under our probabilistic protocol.

**(ii) Non-stationary generalization.** On the volatile Exchange-Rate dataset, Identity-based models (PatchLinear) still achieve the lowest average MSE (Table 19), ruling

*Table 4.* Mann–Whitney U test p-values (one-tailed, $\alpha=0.05$) for Identity Encoder vs. other encoders across all datasets combined.

| Comparison | $p$-value | Significant |
|---|---|---|
| Identity vs. Transformer | $<$**0.0001** | Yes |
| Identity vs. MLP | **0.0115** | Yes |

out benchmark periodicity as a confound.

**(iii) Hyperparameter and seed stability.** Identity-based models maintain consistent optimal configurations across horizons, whereas Transformers require substantially different hyperparameters per horizon on volatile datasets (Section A.6). Multi-seed verification further confirms that these findings are not seed-sensitive (Section A.7).

## 4.2. Auditing Input Transformations under a Unified EC Space

While Experiment 1 focuses on how data is tokenized and processed, this experiment investigates the impact of preprocessing the raw signal via Input Transformations. We designate the Input Transformation ($\mathcal{T}_{in}$) as the Evaluated Object (EO), evaluating three distinct Input Transformations—Trend-Seasonal Decomposition, Multi-Scale Downsampling, and Residual Cycle Forecasting—with RevIN as the shared normalization baseline.

### 4.2.1. EXPERIMENT SETUP

To rigorously audit the utility of Input Transformations, we construct a specialized Evaluation Condition space $\Omega_{\text{priors}}$.

**Evaluated Objects.** We select four representative Input Transformations ($\mathcal{T}_{in}$) to cover the spectrum of structural priors emerging in recent literature. To address distribution shifts and non-stationarity, we include **RevIN** (Kim et al., 2021) as the standard normalization baseline. For temporal decomposition, we examine **Trend-Seasonal Decomposition** (Zeng et al., 2023), which explicitly isolates low-frequency trends. To capture dependencies across varying granularities, we incorporate **Multi-Scale Downsampling** (Wang et al., 2024a), while **Residual Cycle Forecasting** (Lin et al., 2024) is selected to represent explicit periodicity modeling via learnable cycle embeddings.

**Evaluation Conditions.** To rigorously analyze synergy and efficiency, we tailor the Evaluation Condition space around three key dimensions. First, we restrict the backbone embedding to **Point-wise** and **Variate-wise** paradigms, contrasting step-level fragility with channel-level stability. Second, to test the hypothesis that Input Transformations reduce the need for model depth, we expand the capacity search space to include ultra-lightweight configu-

*Table 5.* Performance of 6 Datasets across 4 Horizons on 4 input transformations. Highlighting follows the convention in Table 2. [†]TimeMixer results are $L_{best}$ reported in (Wang et al., 2024a) under their own protocol and are not directly comparable to our paired EC estimates.

| Dataset | BaseLine | | | Cycle | | | MultiScale | | | TrendSeasonal | | | TimeMixer[†] |
|---|---|---|---|---|---|---|---|---|---|---|---|---|---|
| | $\hat{\mu}$ | $\hat{\sigma}$ | $min$ | $\hat{\mu}$ | $\hat{\sigma}$ | $min$ | $\hat{\mu}$ | $\hat{\sigma}$ | $min$ | $\hat{\mu}$ | $\hat{\sigma}$ | $min$ | $min$ |
| ETTh1 | 0.3975 | 0.0315 | 0.3749 | **0.3891** | 0.0283 | **0.3743** | 0.3921 | 0.0183 | 0.3829 | 0.3947 | 0.0248 | 0.3811 | **0.3610** |
| ETTh2 | 0.3013 | 0.0116 | 0.2868 | **0.2969** | 0.0115 | 0.2818 | 0.3071 | 0.0343 | 0.2874 | 0.3024 | 0.0135 | 0.2849 | **0.2710** |
| ETTm1 | 0.3451 | 0.0206 | 0.3139 | 0.3431 | 0.0193 | **0.3137** | **0.3380** | 0.0171 | 0.3159 | 0.3384 | 0.0167 | 0.3166 | **0.2910** |
| ETTm2 | 0.1827 | 0.0037 | 0.1738 | 0.1825 | 0.0019 | **0.1804** | 0.1830 | 0.0011 | 0.1818 | **0.1821** | 0.0013 | 0.1805 | **0.1640** |
| Electricity | 0.2061 | 0.0238 | 0.1679 | 0.1999 | 0.0225 | **0.1580** | **0.1952** | 0.0202 | 0.1580 | 0.1976 | 0.0197 | 0.1729 | **0.1290** |
| Weather | **0.1882** | 0.0094 | **0.1599** | 0.1917 | 0.0017 | 0.1899 | 0.1950 | 0.0021 | 0.1915 | 0.1934 | 0.0008 | 0.1920 | **0.1470** |

rations, sampling hidden dimensions $D \in \{16, \ldots, 512\}$. Finally, we broaden the learning rate spectrum to $\eta \in \{10^{-2}, 10^{-3}, 10^{-4}\}$, specifically assessing whether input simplifications enable stable convergence even under aggressive optimization steps.

### 4.2.2. RESULTS AND ANALYSIS

We analyze the marginalized performance ($\mu$) and stability ($\sigma$) of various Input Transformations ($\mathcal{T}_{in}$) across the evaluated datasets. The comprehensive results are summarized in Table 5; detailed per-embedding and per-encoder breakdowns are provided in Tables 10 and 11 in the Appendix. Our analysis yields three critical insights into the role of Input Transformations:

❶ **Universal Efficacy of Cyclic Priors.** The **Cycle** transformation yields the most consistent gains in both effectiveness ($\mu$) and stability ($\sigma$) across datasets. This confirms that explicit periodic inductive biases are far more efficient than implicit learning, significantly simplifying the optimization landscape even for deep backbones. Notably, this advantage is most pronounced on datasets with strong periodic structure (e.g., Electricity, ETTh1), where the cycle residual effectively removes predictable seasonal variation and allows downstream modules to focus on residual dynamics.

❷ **The Fallacy of Naive Decomposition.** In contrast, naive **Trend-Seasonal** and **Multi-Scale** strategies frequently fail to outperform the **Identity** baseline. A "divide and conquer" approach without subsequent interaction leads to information loss, as isolated branches cannot capture entangled signal dynamics. This finding cautions against treating decomposition as a universally beneficial preprocessing step; its utility depends critically on how the decomposed components are subsequently processed.

❸ **Interaction Defines Utility.** The superior performance of **TimeMixer** over naive Multi-Scale—despite identical inputs—isolates *interaction* as the decisive factor. Input Transformations (like decomposition) are not standalone solutions; they only yield benefits when coupled with architectural mechanisms that enforce cross-component mixing.

*Table 6.* Paired EC evaluation of Identity, time-domain (iTransformer), and frequency-domain (SimpleTM) encoders under Variate-wise Embedding across 6 datasets and 4 prediction horizons. Highlighting follows the convention in Table 2. Detailed results are reported in Table 12.

| Dataset | H | iTransformer | | | SimpleTM | | | Variate and Identity | | |
|---|---|---|---|---|---|---|---|---|---|---|
| | | $\hat{\mu}$ | $\hat{\sigma}$ | $min$ | $\hat{\mu}$ | $\hat{\sigma}$ | $min$ | $\hat{\mu}$ | $\hat{\sigma}$ | $min$ |
| ETTh1 | 96 | 0.3934 | 0.0133 | 0.3789 | 0.3812 | 0.0076 | 0.3730 | 0.3805 | 0.0098 | 0.3641 |
| | 192 | 0.4354 | 0.0099 | 0.4171 | 0.4287 | 0.0105 | 0.4148 | 0.4199 | 0.0179 | 0.3983 |
| | 336 | 0.4656 | 0.0127 | 0.4434 | 0.4601 | 0.0132 | 0.4314 | 0.4583 | 0.0204 | 0.4249 |
| | 720 | 0.4711 | 0.0155 | 0.4504 | 0.4646 | 0.0151 | 0.4473 | 0.4520 | 0.0191 | 0.4366 |
| | avg | 0.4401 | 0.0335 | - | 0.4324 | 0.0354 | - | 0.4271 | 0.0355 | - |
| ETTm2 | 96 | 0.1980 | 0.0291 | 0.1782 | 0.1738 | 0.0061 | 0.1646 | 0.1731 | 0.0079 | 0.1622 |
| | 192 | 0.2478 | 0.0057 | 0.2399 | 0.2332 | 0.0078 | 0.2229 | 0.2296 | 0.0119 | 0.2184 |
| | 336 | 0.3080 | 0.0140 | 0.2894 | 0.2930 | 0.0133 | 0.2735 | 0.2866 | 0.0150 | 0.2707 |
| | 720 | 0.3981 | 0.0195 | 0.3696 | 0.3846 | 0.0170 | 0.3639 | 0.3793 | 0.0184 | 0.3594 |
| | avg | 0.2870 | 0.0803 | - | 0.2697 | 0.0824 | - | 0.2721 | 0.0798 | - |
| Electricity | 96 | 0.1517 | 0.0184 | 0.1335 | 0.1537 | 0.0196 | 0.1322 | 0.1702 | 0.0270 | 0.1428 |
| | 192 | 0.1705 | 0.0145 | 0.1555 | 0.1701 | 0.0190 | 0.1493 | 0.1810 | 0.0184 | 0.1584 |
| | 336 | 0.1930 | 0.0213 | 0.1722 | 0.1929 | 0.0233 | 0.1648 | 0.2015 | 0.0230 | 0.1756 |
| | 720 | 0.2287 | 0.0255 | 0.1972 | 0.2244 | 0.0280 | 0.1964 | 0.2448 | 0.0236 | 0.2137 |
| | avg | 0.1858 | 0.0333 | - | 0.1851 | 0.0329 | - | 0.1974 | 0.0351 | - |

## 4.3. Revisiting Spectral Processing: Performance vs. Necessity

While Experiment 1 identified the dominance of data view (Embedding) and Experiment 2 validated the utility of Input Transformations, a critical question remains regarding the **Encoder** ($\Phi$): Does processing latent representations in the **frequency domain** offer superior effectiveness compared to the **time domain**, and does it also improve stability? Recent works like **SimpleTM** (Chen et al., 2025a) argue that spectral decomposition can effectively disentangle noise from signal. To rigorously verify this, we designate the spectral encoder (from SimpleTM) as the Evaluated Object (EO), alongside the Transformer and Identity encoders from Experiment 1.

### 4.3.1. EXPERIMENT SETUP

**Evaluated Objects.** We adopt a strictly controlled pair to isolate the spectral mechanism. The **Time-Domain** baseline uses **iTransformer** (Variate-wise Embedding + Self-Attention encoder), while the **Frequency-Domain** treatment uses **SimpleTM** (Chen et al., 2025a) with the same Variate-wise Embedding but replaces attention with learnable spectral decomposition (Wavelets/Fourier).

**Evaluation Conditions.** Both encoders are evaluated under the identical EC space from Experiment 1 ($K = 600$ stratified samples across all 6 datasets), ensuring a controlled comparison.

### 4.3.2. RESULTS AND ANALYSIS

The comprehensive comparison across all 6 datasets and 4 prediction horizons is provided in Table 12 in the Appendix.

❶ **Spectral Superiority in Effectiveness.** Under Variate-wise embedding, the **Frequency-Domain** encoder consistently outperforms Time-Domain attention in effectiveness ($\mu$), yet exhibits comparable stability ($\sigma$). This indicates that spectral decomposition improves signal representation rather than acting as a stabilizing filter against hyperparameter sensitivity.

❷ **The "Identity" Ceiling.** Crucially, spectral processing fails to surpass the parameter-free **Identity** baseline on datasets such as ETTh1 and ETTm2, implying that its effectiveness gain over attention stems merely from introducing *less signal distortion* rather than extracting meaningful patterns beyond what the optimized data view already provides.

## 5. Limitations and Future Work

Our evaluation currently covers six widely used benchmarks with a fixed set of modular alternatives; expanding to more diverse datasets (e.g., longer horizons, higher-dimensional variates) and emerging architectures would strengthen the generality of our conclusions. In addition, the current modular decomposition assumes clear boundaries between pipeline stages, whereas some recent models blur these boundaries through end-to-end training or shared parameters across stages, posing challenges for clean attribution. The probabilistic evaluation protocol also relies on a fixed EC sampling strategy; adaptive or data-dependent sampling strategies could reveal different aspects of model behavior. Future work should further investigate how intrinsic data characteristics—such as periodicity, noise level, nonstationarity, and cross-variate dependency—interact with modular choices, and develop automated composition strategies that select or assemble suitable Input Transformations, Embeddings, and Encoders according to these data properties.

## 6. Conclusion

This paper introduced **CombinationTS**, a modular and probabilistic framework that shifts TSF evaluation from monolithic model selection to component-level attribution under a shared Evaluation Condition space. Large-scale paired evaluation shows that modern forecasting gains are often driven more by data view and structural priors than by complex encoders: a well-designed Embedding can make a parameter-free **Identity** Encoder competitive with deep alternatives, Input Transformations can improve the effectiveness–stability trade-off when properly coupled with interactions, and spectral processing improves marginalized performance without substantially reducing instability. These findings call for future TSF architectures to demonstrate consistent gains in both **Marginalized Performance** ($\hat{\mu}$) and **Stability** ($\hat{\sigma}$) over strong modular baselines, rather than relying on fragile leaderboard peaks.

## Acknowledgements

Supported by the Innovation Funding of ICT, CAS under Grant No. E661110.

## Impact Statement

This paper presents CombinationTS, a modular evaluation framework for time series forecasting, with the goal of advancing methodological rigor and interpretability in machine learning research. The work focuses on improving how forecasting models are analyzed and compared, rather than introducing new predictive capabilities.

The proposed framework is intended for benign applications such as scientific analysis, industrial forecasting, and infrastructure-related time series modeling, where reliable evaluation and reproducibility are essential. By emphasizing modular attribution and distributional evaluation, this work may help reduce misattributed performance gains and encourage more robust and efficient model design.

We do not anticipate negative societal impacts beyond those commonly associated with machine learning research, such as potential misuse or over-reliance on automated predictions. These risks are not specific to this work and can be mitigated through responsible use and appropriate human oversight. We do not identify ethical concerns requiring special consideration beyond standard practices in the field.

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

# A. Implementation Details

## A.1. Dataset Descriptions

We evaluate our approach on **six** widely used real-world multivariate time series datasets. These datasets originate from diverse application domains, including electricity systems, meteorology, traffic dynamics, and energy consumption, which enables a comprehensive assessment of the robustness and general applicability of the proposed method. The overall statistics of the datasets are summarized in Table 7. Specifically, the datasets are described as follows:

- **Electricity Transformer Temperature (ETT)** (Zhou et al., 2021): This benchmark consists of four subsets. ETTh1 and ETTh2 are sampled at an hourly frequency, while ETTm1 and ETTm2 are collected at 15-minute intervals. All time series are measured from two electricity transformers.

- **Weather** (Wu et al., 2021): This dataset contains 21 meteorological variables collected at 10-minute intervals from a weather station in Germany during 2020, released by the Max Planck Institute for Biogeochemistry.

- **Electricity** (Wu et al., 2021): This dataset records hourly electricity consumption from 321 individual clients over the period from 2012 to 2014.

*Table 7.* Dataset statistics. "Var" denotes the number of variates; "Dataset Size" shows the number of time points in (Train/Validation/Test) splits; "Frequency" is the sampling interval.

| Dataset | Var | Dataset Size (Train/Val/Test) | Frequency |
|---|---|---|---|
| ETTh1, ETTh2 | 7 | (8545, 2881, 2881) | Hourly |
| ETTm1, ETTm2 | 7 | (34465, 11521, 11521) | 15 min |
| Weather | 21 | (36792, 5271, 10540) | 10 min |
| Electricity | 321 | (18317, 2633, 5261) | Hourly |

## A.2. Evaluation Metrics

We follow standard evaluation protocols in time series forecasting and adopt commonly used metrics to assess predictive performance across all datasets.

For the considered benchmarks, we adopt Mean Squared Error (MSE) as the primary metric, defined as:

- Mean Squared Error (MSE): $\frac{1}{T} \sum_{t=1}^{T} (\hat{y}_t - y_t)^2$

This metric is widely adopted in prior studies (Nie et al., 2023) and penalizes large deviations more strongly, providing a rigorous measure of forecasting accuracy. All metrics are averaged over all variates and prediction horizons, with lower values indicating better performance.

## A.3. Experiment details

All models were implemented in PyTorch and trained on a Linux server with NVIDIA V100 GPUs.

**Hyperparameter Search Space.** During the hyperparameter optimization process, all methods share the same predefined search space to guarantee fairness. Specifically, the candidate ranges are defined as follows:

- `Look-back windows` $\in \{96, 192, 336, 512\}$,

- `Prediction horizons` $\in \{96, 192, 336, 720\}$,

- `batch size` $\in \{32\}$,

- `epochs` $\in \{30\}$,

- `early stopping patience` $\in \{3\}$,

*Table 8.* Experiment 1 (Dissecting the Backbone): Performance of 6 Datasets across 4 Horizons on 6 Embeddings. Highlighting follows the convention in Table 2.

| Dataset | H | Point wise | | | Patch wise | | | Variate wise | | | Identity | | | L to dim | | | C to dim | | |
|---|---|---|---|---|---|---|---|---|---|---|---|---|---|---|---|---|---|---|---|
| | | $\hat{\mu}$ | $\hat{\sigma}$ | $min$ | $\hat{\mu}$ | $\hat{\sigma}$ | $min$ | $\hat{\mu}$ | $\hat{\sigma}$ | $min$ | $\hat{\mu}$ | $\hat{\sigma}$ | $min$ | $\hat{\mu}$ | $\hat{\sigma}$ | $min$ | $\hat{\mu}$ | $\hat{\sigma}$ | $min$ |
| ETTh1 | 96 | 0.3898 | 0.0143 | 0.3651 | 0.3772 | 0.0058 | 0.3645 | 0.3834 | 0.0080 | 0.3641 | 0.5890 | 0.1537 | 0.3721 | 0.3937 | 0.0156 | 0.3721 | 0.4602 | 0.0357 | 0.4013 |
| | 192 | 0.4324 | 0.0166 | 0.3992 | 0.4240 | 0.0133 | 0.3984 | 0.4247 | 0.0128 | 0.3983 | 0.6092 | 0.1408 | 0.4028 | 0.4343 | 0.0161 | 0.4028 | 0.5096 | 0.0342 | 0.4484 |
| | 336 | 0.4651 | 0.0193 | 0.4255 | 0.4575 | 0.0170 | 0.4196 | 0.4589 | 0.0148 | 0.4249 | 0.6262 | 0.1209 | 0.4294 | 0.4693 | 0.0160 | 0.4294 | 0.5219 | 0.0175 | 0.4859 |
| | 720 | 0.4772 | 0.0415 | 0.4248 | 0.4812 | 0.0326 | 0.4366 | 0.4618 | 0.0140 | 0.4366 | 0.6317 | 0.1305 | 0.4409 | 0.4751 | 0.0223 | 0.4409 | 0.5652 | 0.0474 | 0.4934 |
| | avg | 0.4367 | 0.0364 | - | 0.4277 | 0.0370 | - | 0.4285 | 0.0327 | - | 0.6133 | 0.1362 | - | 0.4398 | 0.0342 | - | 0.5104 | 0.0451 | - |
| ETTh2 | 96 | 0.3076 | 0.0264 | 0.2684 | 0.2912 | 0.0108 | 0.2699 | 0.2963 | 0.0163 | 0.2656 | 0.3438 | 0.0444 | 0.2751 | 0.3034 | 0.0171 | 0.2751 | 0.3874 | 0.0245 | 0.3552 |
| | 192 | 0.3799 | 0.0267 | 0.3258 | 0.3626 | 0.0153 | 0.3265 | 0.3709 | 0.0189 | 0.3264 | 0.3969 | 0.0293 | 0.3363 | 0.3752 | 0.0184 | 0.3363 | 0.4380 | 0.0151 | 0.3946 |
| | 336 | 0.4123 | 0.0286 | 0.3502 | 0.3949 | 0.0202 | 0.3502 | 0.3937 | 0.0191 | 0.3509 | 0.4102 | 0.0278 | 0.3553 | 0.3998 | 0.0204 | 0.3553 | 0.4478 | 0.0201 | 0.4147 |
| | 720 | 0.4365 | 0.0363 | 0.3799 | 0.4205 | 0.0178 | 0.3852 | 0.4137 | 0.0140 | 0.3841 | 0.4408 | 0.0352 | 0.3897 | 0.4203 | 0.0174 | 0.3897 | 0.4731 | 0.0168 | 0.4481 |
| | avg | 0.3838 | 0.0513 | - | 0.3657 | 0.0459 | - | 0.3689 | 0.0436 | - | 0.3956 | 0.0416 | - | 0.3751 | 0.0436 | - | 0.4376 | 0.0330 | - |
| ETTm1 | 96 | 0.3134 | 0.0136 | 0.2866 | 0.3054 | 0.0141 | 0.2837 | 0.3117 | 0.0142 | 0.2878 | 0.5506 | 0.1744 | 0.3030 | 0.3212 | 0.0171 | 0.3012 | 0.3437 | 0.0222 | 0.3085 |
| | 192 | 0.3558 | 0.0172 | 0.3298 | 0.3483 | 0.0188 | 0.3235 | 0.3515 | 0.0186 | 0.3280 | 0.5708 | 0.1613 | 0.3368 | 0.3596 | 0.0194 | 0.3368 | 0.3883 | 0.0198 | 0.3471 |
| | 336 | 0.3889 | 0.0172 | 0.3650 | 0.3776 | 0.0145 | 0.3581 | 0.3866 | 0.0210 | 0.3615 | 0.5908 | 0.1522 | 0.3714 | 0.3923 | 0.0180 | 0.3714 | 0.4242 | 0.0213 | 0.3860 |
| | 720 | 0.4391 | 0.0146 | 0.4129 | 0.4309 | 0.0126 | 0.4128 | 0.4393 | 0.0160 | 0.4179 | 0.6190 | 0.1357 | 0.4228 | 0.4469 | 0.0180 | 0.4228 | 0.4758 | 0.0225 | 0.4417 |
| | avg | 0.3686 | 0.0458 | - | 0.3603 | 0.0459 | - | 0.3676 | 0.0478 | - | 0.5821 | 0.1569 | - | 0.3740 | 0.0458 | - | 0.4026 | 0.0507 | - |
| ETTm2 | 96 | 0.1800 | 0.0089 | 0.1625 | 0.1728 | 0.0051 | 0.1631 | 0.1758 | 0.0072 | 0.1622 | 0.2235 | 0.0416 | 0.1633 | 0.1779 | 0.0073 | 0.1633 | 0.1993 | 0.0146 | 0.1779 |
| | 192 | 0.2414 | 0.0136 | 0.2170 | 0.2331 | 0.0079 | 0.2179 | 0.2343 | 0.0106 | 0.2184 | 0.2739 | 0.0345 | 0.2180 | 0.2370 | 0.0115 | 0.2180 | 0.2639 | 0.0135 | 0.2406 |
| | 336 | 0.3019 | 0.0172 | 0.2701 | 0.2911 | 0.0105 | 0.2732 | 0.2928 | 0.0135 | 0.2707 | 0.3218 | 0.0265 | 0.2711 | 0.2968 | 0.0134 | 0.2711 | 0.3280 | 0.0121 | 0.3099 |
| | 720 | 0.3981 | 0.0216 | 0.3591 | 0.3813 | 0.0140 | 0.3608 | 0.3863 | 0.0170 | 0.3594 | 0.4095 | 0.0236 | 0.3603 | 0.3892 | 0.0174 | 0.3603 | 0.4368 | 0.0145 | 0.4138 |
| | avg | 0.2663 | 0.0762 | - | 0.2562 | 0.0725 | - | 0.2596 | 0.0723 | - | 0.2975 | 0.0704 | - | 0.2612 | 0.0728 | - | 0.2937 | 0.0827 | - |
| Electricity | 96 | 0.1609 | 0.0228 | 0.1431 | 0.1512 | 0.0176 | 0.1315 | 0.1527 | 0.0135 | 0.1335 | 0.1639 | 0.0193 | 0.1440 | 0.1573 | 0.0190 | 0.1349 | 0.1877 | 0.0106 | 0.1658 |
| | 192 | 0.1716 | 0.0154 | 0.1574 | 0.1641 | 0.0111 | 0.1465 | 0.1710 | 0.0111 | 0.1555 | 0.1777 | 0.0169 | 0.1583 | 0.1710 | 0.0115 | 0.1542 | - | - | - |
| | 336 | 0.1917 | 0.0170 | 0.1735 | 0.1828 | 0.0137 | 0.1644 | 0.1906 | 0.0153 | 0.1722 | 0.2270 | 0.1270 | 0.1748 | 0.1928 | 0.0170 | 0.1706 | - | - | - |
| | 720 | 0.2342 | 0.0194 | 0.2132 | 0.2236 | 0.0160 | 0.2004 | 0.2293 | 0.0205 | 0.1972 | 0.2335 | 0.0195 | 0.2144 | 0.2218 | 0.0141 | 0.1991 | - | - | - |
| | avg | 0.1855 | 0.0252 | - | 0.1773 | 0.0246 | - | 0.1827 | 0.0249 | - | 0.2053 | 0.0730 | - | 0.1851 | 0.0264 | - | 0.1877 | 0.0106 | - |
| Weather | 96 | 0.1715 | 0.0114 | 0.1543 | 0.1673 | 0.0135 | 0.1474 | 0.1679 | 0.0126 | 0.1479 | 0.2120 | 0.0279 | 0.1718 | 0.1705 | 0.0106 | 0.1468 | 0.1741 | 0.0132 | 0.1587 |
| | 192 | 0.2182 | 0.0120 | 0.1969 | 0.2142 | 0.0127 | 0.1912 | 0.2144 | 0.0129 | 0.1914 | 0.2519 | 0.0205 | 0.2162 | 0.2147 | 0.0096 | 0.1961 | 0.2260 | 0.0150 | 0.2090 |
| | 336 | 0.2639 | 0.0107 | 0.2473 | 0.2596 | 0.0121 | 0.2422 | 0.2593 | 0.0109 | 0.2415 | 0.2950 | 0.0215 | 0.2615 | 0.2610 | 0.0105 | 0.2454 | 0.2817 | 0.0137 | 0.2644 |
| | 720 | 0.3391 | 0.0118 | 0.3180 | 0.3357 | 0.0134 | 0.3158 | 0.3345 | 0.0135 | 0.3131 | 0.3594 | 0.0168 | 0.3267 | 0.3386 | 0.0124 | 0.3197 | 0.3652 | 0.0194 | 0.3421 |
| | avg | 0.2299 | 0.0514 | - | 0.2271 | 0.0526 | - | 0.2274 | 0.0521 | - | 0.2655 | 0.0502 | - | 0.2287 | 0.0520 | - | 0.2436 | 0.0610 | - |

- learning rate $\in \{1\text{e-}3, 1\text{e-}4\}$,

- encoder layers $\in \{1, 2, 3\}$,

- hidden dimensions $\in \{64, 128, 256, 512\}$

All baselines are evaluated within a common search space that encompasses architectural design choices and optimization-related settings, and this protocol is applied consistently across methods. Consequently, the reported performance better reflects the intrinsic representational capability of each model, instead of discrepancies induced by unequal hyperparameter tuning.

## A.4. Statistical Significance Analysis

The overall Mann–Whitney U test results are reported in Table 4 in the main text. Here we provide the per-dataset breakdown. As shown in Table 18, Identity is significantly better than Transformer in 3/6 datasets and MLP in 2/6 datasets ($p < 0.05$), while never being significantly worse on any dataset.

## A.5. Generalizability to Non-Stationary Domains

To assess whether the Identity Paradox extends beyond the six standard benchmarks (which are dominated by periodic, low-noise signals), we evaluate on the **Exchange-Rate** dataset, a widely used non-stationary benchmark with volatile, non-periodic dynamics. We conduct a targeted hyperparameter search for two representative Identity-based

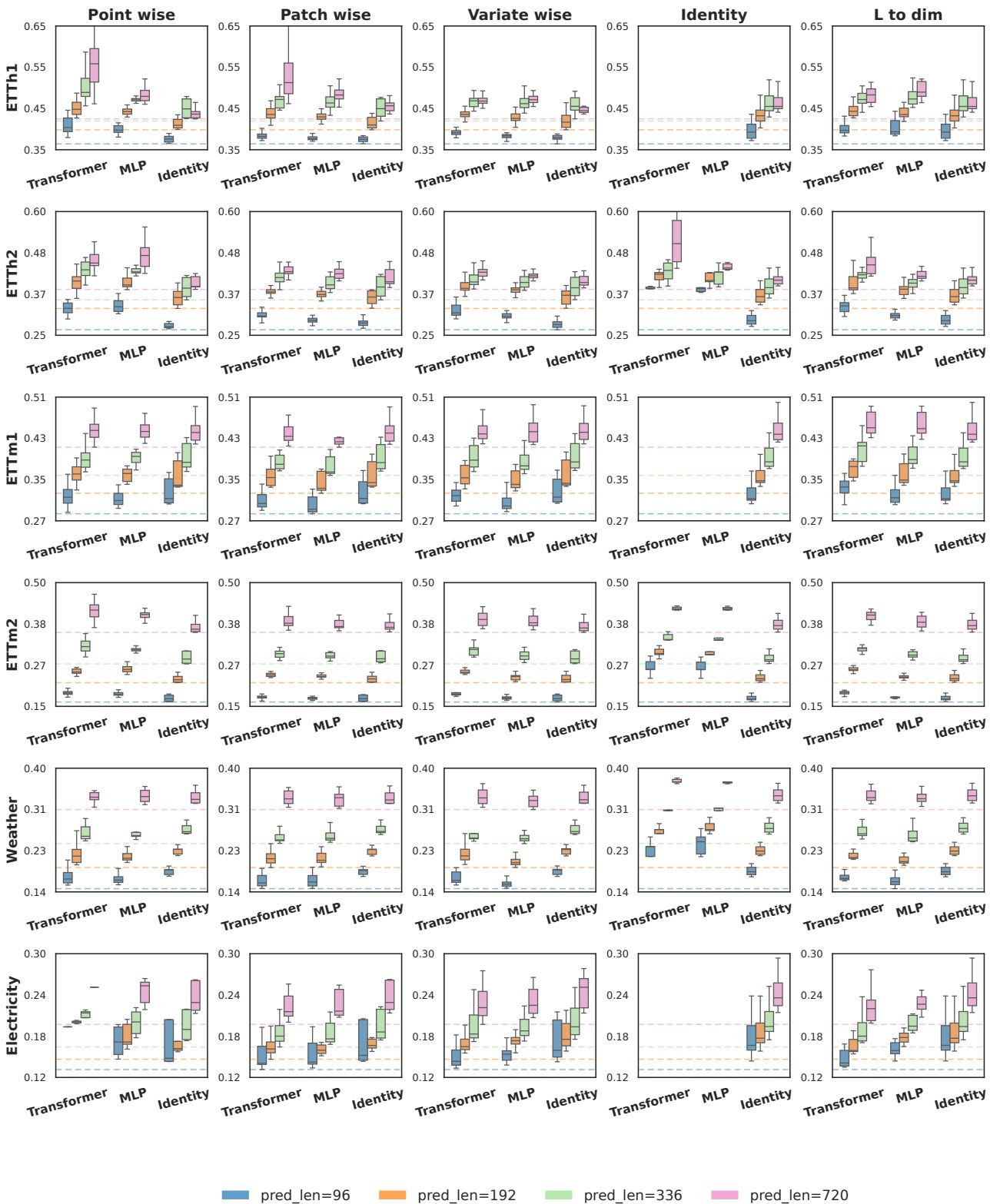

*Figure 4.* Distribution of encoder effectiveness under paired EC sampling (full figure).

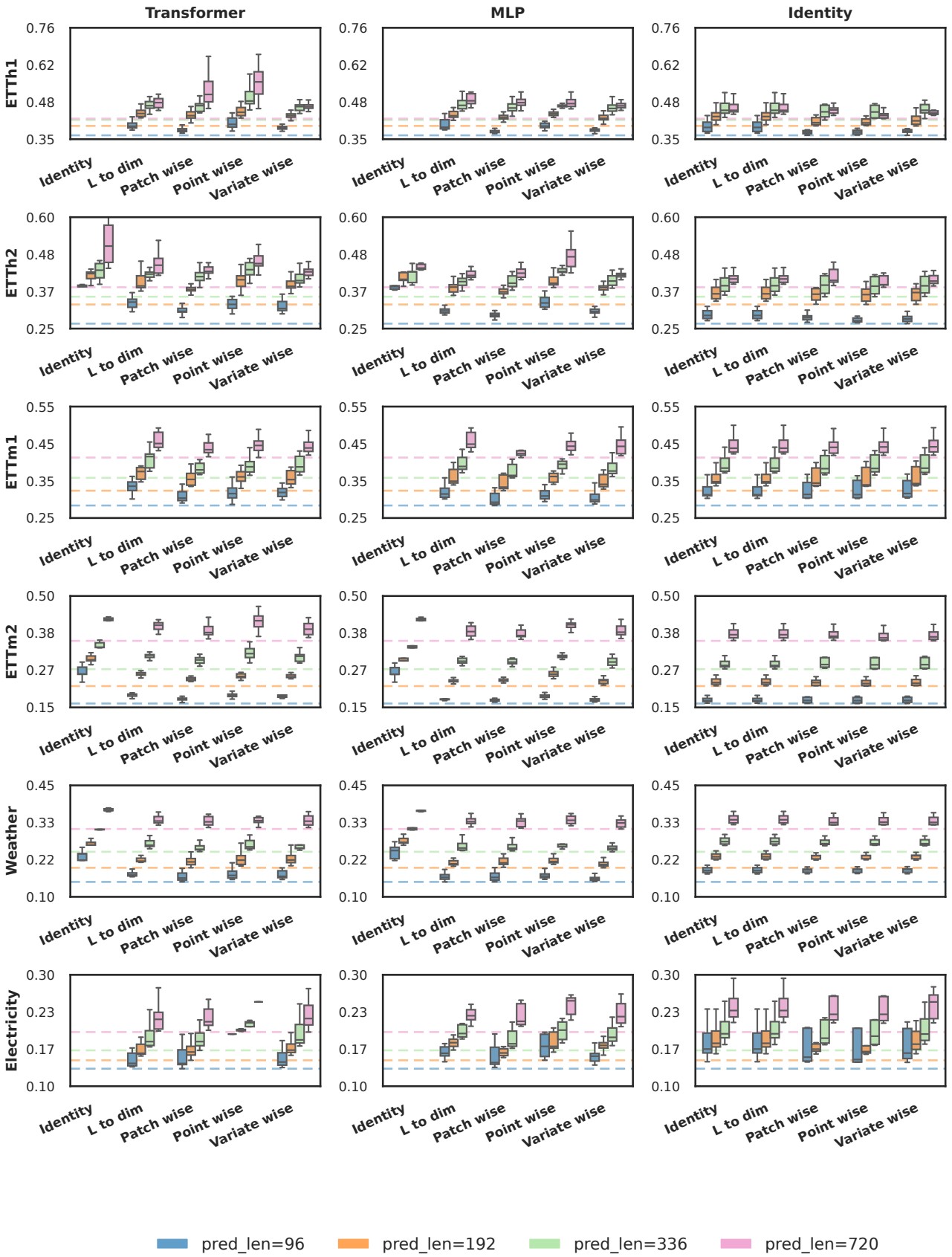

*Figure 5.* Distribution of embedding effectiveness under paired EC sampling (full figure).

models—**PatchLinear** (Patch-wise Embedding + Identity Encoder) and **iLinear** (Variate-wise Embedding + Identity Encoder)—and compare against iTransformer and PatchTST under their respective published configurations.

As shown in Table 19, PatchLinear achieves the lowest average MSE (0.357) across all prediction horizons, outperforming both iTransformer (0.360) and PatchTST (0.367). This confirms that the Identity Paradox is not an artifact of benchmark periodicity: even on a non-stationary, volatile dataset, Identity-based models with well-designed embeddings remain competitive, reinforcing the dominance of data view over Encoder complexity.

### A.6. Hyperparameter Tunability Analysis

A key concern regarding the Identity Paradox is whether the observed performance advantage stems from favorable default hyperparameters rather than intrinsic architectural simplicity. To address this, we analyze the **optimal hyperparameter configurations** identified by our paired EC sampling for representative models on two contrasting datasets.

**Robust Case: Electricity.** On Electricity, both PatchLinear (Identity Encoder) and PatchTST (Transformer Encoder) converge to remarkably consistent optimal configurations across all prediction horizons (Table 20). Specifically, both models consistently select $seq\_len = 512$ and $\eta = 0.001$, indicating that on datasets with strong periodicity and ample training data, the hyperparameter landscape is smooth and model-agnostic.

**Brittle Case: ETTm1.** On ETTm1, the contrast is stark (Table 21). While PatchLinear maintains a stable configuration ($seq\_len = 512$, $\eta = 0.001$), PatchTST's optimal $d_{model}$ fluctuates substantially ($64 \rightarrow 256 \rightarrow 128$) and the learning rate oscillates between 0.001 and 0.0001. This brittleness confirms that the Transformer's performance is highly sensitive to hyperparameter choices, while the Identity Encoder's simplicity inherently avoids such tuning instability.

### A.7. Multi-Seed Robustness Verification

To verify that the Identity Paradox is not an artifact of a specific random seed, we repeat the full paired EC evaluation on ETTh1 using three independent seeds (333, 2025, 2026) for both PatchLinear (Patch-wise Embedding + Identity Encoder) and PatchTST (Patch-wise Embedding + Transformer Encoder).

As shown in Table 22, both models exhibit negligible variation in $\hat{\mu}$ (within 0.003) and moderate variation in $\hat{\sigma}$ across the three seeds. Crucially, the relative ordering—PatchLinear outperforming PatchTST in both effectiveness and stability—is preserved across all seeds, confirming that the Identity Paradox is a robust finding rather than a seed-dependent artifact.

*Table 9.* Experiment 1 (Dissecting the Backbone): Performance of 6 Datasets across 4 Horizons on 3 Encoders. Highlighting follows the convention in Table 2.

| Dataset | H | MLP | | | Transformer | | | Identity | | |
|---|---|---|---|---|---|---|---|---|---|---|
| | | $\hat{\mu}$ | $\hat{\sigma}$ | $min$ | $\hat{\mu}$ | $\hat{\sigma}$ | $min$ | $\hat{\mu}$ | $\hat{\sigma}$ | $min$ |
| ETTh1 | 96 | **0.4371** | 0.0816 | **0.3700** | 0.4518 | 0.0930 | 0.3721 | **0.3907** | 0.0231 | **0.3641** |
| | 192 | **0.4870** | 0.0827 | **0.4043** | 0.5044 | 0.0988 | 0.4091 | **0.4334** | 0.0306 | **0.3983** |
| | 336 | **0.5186** | 0.0787 | **0.4335** | 0.5239 | 0.0795 | 0.4406 | **0.4684** | 0.0268 | **0.4196** |
| | 720 | **0.5538** | 0.1039 | 0.4533 | 0.5679 | 0.1014 | **0.4504** | **0.4722** | 0.0406 | **0.4248** |
| | avg | **0.4960** | 0.0938 | - | 0.5095 | 0.1004 | - | **0.4392** | 0.0424 | - |
| ETTh2 | 96 | **0.3373** | 0.0423 | **0.2770** | 0.3498 | 0.0437 | 0.2835 | **0.3014** | 0.0368 | **0.2656** |
| | 192 | **0.3985** | 0.0293 | **0.3468** | 0.4060 | 0.0310 | 0.3485 | **0.3753** | 0.0373 | **0.3258** |
| | 336 | **0.4197** | 0.0288 | **0.3680** | 0.4317 | 0.0308 | 0.3788 | **0.3999** | 0.0325 | **0.3502** |
| | 720 | **0.4460** | 0.0339 | **0.4034** | 0.4616 | 0.0378 | 0.4065 | **0.4231** | 0.0374 | **0.3799** |
| | avg | **0.4009** | 0.0473 | - | 0.4135 | 0.0512 | - | **0.3778** | 0.0547 | - |
| ETTm1 | 96 | **0.3414** | 0.0785 | **0.2837** | 0.3704 | 0.0956 | **0.2866** | **0.3192** | 0.0185 | 0.3026 |
| | 192 | **0.3863** | 0.0695 | **0.3235** | 0.4103 | 0.0890 | **0.3298** | **0.3593** | 0.0221 | 0.3354 |
| | 336 | **0.4235** | 0.0734 | **0.3581** | 0.4435 | 0.0876 | **0.3650** | **0.3919** | 0.0199 | 0.3658 |
| | 720 | **0.4810** | 0.0697 | **0.4128** | 0.4930 | 0.0830 | **0.4129** | **0.4408** | 0.0158 | 0.4185 |
| | avg | **0.4079** | 0.0912 | - | 0.4282 | 0.0997 | - | **0.3721** | 0.0457 | - |
| ETTm2 | 96 | **0.1893** | 0.0197 | 0.1667 | 0.2027 | 0.0265 | **0.1652** | **0.1755** | 0.0087 | **0.1622** |
| | 192 | **0.2525** | 0.0213 | **0.2201** | 0.2610 | 0.0221 | 0.2313 | **0.2341** | 0.0133 | **0.2170** |
| | 336 | **0.3101** | 0.0198 | **0.2738** | 0.3202 | 0.0199 | 0.2794 | **0.2956** | 0.0194 | **0.2701** |
| | 720 | **0.4057** | 0.0232 | **0.3634** | 0.4163 | 0.0265 | 0.3653 | **0.3919** | 0.0270 | **0.3591** |
| | avg | **0.2784** | 0.0759 | - | 0.2909 | 0.0772 | - | **0.2621** | 0.0756 | - |
| Electricity | 96 | **0.1599** | 0.0171 | **0.1338** | **0.1535** | 0.0194 | **0.1315** | 0.1679 | 0.0254 | 0.1428 |
| | 192 | **0.1729** | 0.0130 | **0.1515** | **0.1671** | 0.0129 | **0.1465** | 0.1742 | 0.0161 | 0.1574 |
| | 336 | **0.1925** | 0.0158 | **0.1681** | **0.1858** | 0.0153 | **0.1644** | 0.1939 | 0.0179 | 0.1735 |
| | 720 | **0.2282** | 0.0171 | **0.2070** | **0.2196** | 0.0171 | **0.1972** | 0.2326 | 0.0191 | 0.2132 |
| | avg | **0.1851** | 0.0253 | - | **0.1776** | 0.0242 | - | 0.1888 | 0.0275 | - |
| Weather | 96 | **0.1714** | 0.0180 | **0.1468** | **0.1763** | 0.0198 | **0.1474** | 0.1767 | 0.0101 | 0.1587 |
| | 192 | **0.2191** | 0.0192 | **0.1914** | 0.2289 | 0.0235 | **0.1912** | **0.2238** | 0.0087 | 0.2090 |
| | 336 | **0.2717** | 0.0224 | **0.2415** | 0.2727 | 0.0237 | **0.2422** | **0.2714** | 0.0084 | 0.2605 |
| | 720 | **0.3461** | 0.0192 | **0.3131** | 0.3502 | 0.0209 | **0.3179** | **0.3436** | 0.0129 | 0.3257 |
| | avg | **0.2360** | 0.0569 | - | 0.2424 | 0.0588 | - | **0.2348** | 0.0514 | - |

*Table 10.* Experiment 2 (Auditing Input Transformations): Ablation study on embedding strategies: Performance comparison between channel-wise and point-wise embeddings across 6 datasets. Highlighting follows the convention in Table 2.

| Dataset | Channel | | | Point | | |
|---|---|---|---|---|---|---|
| | $\hat{\mu}$ | $\hat{\sigma}$ | $min$ | $\hat{\mu}$ | $\hat{\sigma}$ | $min$ |
| **ETTh1** | 0.4028 | 0.0369 | **0.3743** | **0.3866** | 0.0090 | 0.3747 |
| **ETTh2** | **0.3010** | 0.0139 | **0.2818** | 0.3028 | 0.0253 | 0.2829 |
| **ETTm1** | 0.3462 | 0.0186 | 0.3160 | **0.3362** | 0.0173 | **0.3137** |
| **ETTm2** | 0.1830 | 0.0035 | **0.1738** | **0.1822** | 0.0018 | 0.1778 |
| **Electricity** | 0.2018 | 0.0291 | **0.1580** | **0.1976** | 0.0102 | 0.1840 |
| **Weather** | **0.1903** | 0.0096 | **0.1599** | 0.1913 | 0.0038 | 0.1816 |

*Table 11.* Experiment 2 (Auditing Input Transformations): Ablation study on encoder architectures: Comparing Identity, MLP, and Transformer-based encoders across six datasets. Highlighting follows the convention in Table 2.

| Dataset | Identity | | | MLP | | | Transformer | | |
|---|---|---|---|---|---|---|---|---|---|
| | $\hat{\mu}$ | $\hat{\sigma}$ | $min$ | $\hat{\mu}$ | $\hat{\sigma}$ | $min$ | $\hat{\mu}$ | $\hat{\sigma}$ | $min$ |
| **ETTh1** | **0.3932** | 0.0253 | **0.3743** | **0.3971** | 0.0334 | **0.3749** | 0.3984 | 0.0330 | 0.3773 |
| **ETTh2** | **0.2908** | 0.0091 | **0.2829** | **0.3052** | 0.0123 | **0.2818** | 0.3097 | 0.0287 | 0.2873 |
| **ETTm1** | 0.3613 | 0.0072 | 0.3522 | **0.3314** | 0.0123 | **0.3159** | **0.3308** | 0.0152 | **0.3137** |
| **ETTm2** | **0.1827** | 0.0017 | 0.1804 | **0.1805** | 0.0042 | **0.1738** | 0.1843 | 0.0034 | **0.1770** |
| **Electricity** | 0.2078 | 0.0152 | 0.1981 | **0.1930** | 0.0217 | **0.1588** | **0.1983** | 0.0251 | **0.1580** |
| **Weather** | 0.1935 | 0.0019 | 0.1899 | **0.1807** | 0.0114 | **0.1599** | **0.1901** | 0.0069 | **0.1785** |

*Table 12.* Experiment 3 (Revisiting Spectral Processing): Comprehensive performance evaluation across various prediction horizons: A comparison between iTransformer and SimpleTM. Highlighting follows the convention in Table 2.

| Dataset | H | iTransformer | | | SimpleTM | | |
|---|---|---|---|---|---|---|---|
| | | $\hat{\mu}$ | $\hat{\sigma}$ | $min$ | $\hat{\mu}$ | $\hat{\sigma}$ | $min$ |
| **ETTh1** | 96 | 0.3934 | 0.0133 | 0.3789 | **0.3812** | 0.0076 | **0.3730** |
| | 192 | 0.4354 | 0.0099 | 0.4171 | **0.4287** | 0.0105 | **0.4148** |
| | 336 | 0.4656 | 0.0127 | 0.4434 | **0.4601** | 0.0132 | **0.4314** |
| | 720 | 0.4711 | 0.0155 | 0.4504 | **0.4646** | 0.0151 | **0.4473** |
| | avg | 0.4401 | 0.0335 | - | **0.4324** | 0.0354 | - |
| **ETTh2** | 96 | 0.3214 | 0.0228 | 0.2964 | **0.2945** | 0.0065 | **0.2840** |
| | 192 | 0.3887 | 0.0167 | 0.3599 | **0.3709** | 0.0110 | **0.3520** |
| | 336 | 0.4076 | 0.0209 | 0.3809 | **0.4030** | 0.0204 | **0.3727** |
| | 720 | 0.4309 | 0.0227 | 0.4065 | **0.4148** | 0.0120 | **0.3921** |
| | avg | 0.3894 | 0.0434 | - | **0.3736** | 0.0463 | - |
| **ETTm1** | 96 | 0.3507 | 0.1022 | 0.2987 | **0.2974** | 0.0164 | **0.2825** |
| | 192 | 0.3670 | 0.0519 | 0.3319 | **0.3454** | 0.0186 | **0.3232** |
| | 336 | 0.4044 | 0.0608 | 0.3655 | **0.3792** | 0.0179 | **0.3590** |
| | 720 | 0.4447 | 0.0194 | 0.4194 | **0.4334** | 0.0171 | **0.4183** |
| | avg | 0.3911 | 0.0764 | - | **0.3624** | 0.0551 | - |
| **ETTm2** | 96 | 0.1980 | 0.0291 | 0.1782 | **0.1738** | 0.0061 | **0.1646** |
| | 192 | 0.2478 | 0.0057 | 0.2399 | **0.2332** | 0.0078 | **0.2229** |
| | 336 | 0.3080 | 0.0140 | 0.2894 | **0.2930** | 0.0133 | **0.2735** |
| | 720 | 0.3981 | 0.0195 | 0.3696 | **0.3846** | 0.0170 | **0.3639** |
| | avg | 0.2870 | 0.0803 | - | **0.2697** | 0.0824 | - |
| **Electricity** | 96 | **0.1517** | 0.0184 | 0.1335 | 0.1537 | 0.0196 | **0.1322** |
| | 192 | 0.1705 | 0.0145 | 0.1555 | **0.1701** | 0.0160 | **0.1493** |
| | 336 | 0.1930 | 0.0213 | 0.1722 | **0.1929** | 0.0233 | **0.1648** |
| | 720 | 0.2287 | 0.0255 | 0.1972 | **0.2244** | 0.0280 | **0.1964** |
| | avg | 0.1858 | 0.0333 | - | **0.1851** | 0.0329 | - |
| **Weather** | 96 | 0.1738 | 0.0231 | 0.1544 | **0.1564** | 0.0118 | **0.1440** |
| | 192 | 0.2230 | 0.0226 | 0.1976 | **0.2030** | 0.0121 | **0.1873** |
| | 336 | 0.2610 | 0.0156 | 0.2465 | **0.2524** | 0.0133 | **0.2376** |
| | 720 | 0.3384 | 0.0157 | 0.3179 | **0.3305** | 0.0145 | **0.3137** |
| | avg | 0.2412 | 0.0621 | - | **0.2274** | 0.0644 | - |

*Table 13.* Experiment 2 (Auditing Input Transformations): Performance of 6 Datasets across 4 Horizons on 4 Input Transformations. Highlighting follows the convention in Table 2, where red denotes MSE and blue denotes MAE.

| Methods | | Comparison Baseline | | | | | | | | Compared Model | | | | | | |
|---|---|---|---|---|---|---|---|---|---|---|---|---|---|---|---|---|
| | | PatchLinear (Ours) | | iLinear (Ours) | | PointLinear (Ours) | | RLinear (2021) | | TimeMixer (2024a) | | SimpleTM (2025a) | | PatchTST (2023) | | iTransformer (2024) |
| Metric | | MSE | MAE | MSE | MAE | MSE | MAE | MSE | MAE | MSE | MAE | MSE | MAE | MSE | MAE | MSE | MAE |
| **ETTh1** | 96 | 0.365 | 0.392 | **0.364** | 0.392 | 0.365 | 0.392 | 0.366 | **0.391** | **0.361** | **0.390** | 0.366 | 0.392 | 0.370 | 0.400 | 0.386 | 0.405 |
| | 192 | **0.398** | 0.414 | **0.398** | 0.414 | **0.399** | **0.413** | 0.404 | **0.412** | 0.409 | 0.414 | 0.422 | 0.421 | 0.413 | 0.429 | 0.441 | 0.436 |
| | 336 | **0.420** | **0.429** | 0.425 | 0.435 | 0.426 | 0.432 | **0.420** | **0.423** | 0.430 | **0.429** | 0.440 | 0.438 | **0.422** | 0.440 | 0.487 | 0.458 |
| | 720 | **0.437** | 0.459 | **0.437** | 0.460 | **0.425** | **0.451** | 0.442 | **0.456** | 0.445 | 0.460 | 0.463 | 0.462 | 0.447 | 0.468 | 0.503 | 0.491 |
| | avg | **0.405** | 0.424 | 0.406 | 0.425 | **0.404** | **0.422** | 0.408 | **0.421** | 0.411 | 0.423 | 0.423 | 0.428 | 0.413 | 0.434 | 0.454 | 0.448 |
| **ETTh2** | 96 | 0.270 | 0.337 | **0.266** | 0.334 | 0.268 | 0.335 | **0.262** | **0.331** | 0.271 | **0.330** | 0.281 | 0.338 | 0.274 | 0.337 | 0.297 | 0.349 |
| | 192 | 0.327 | 0.376 | 0.326 | **0.373** | 0.326 | **0.372** | **0.319** | 0.374 | **0.317** | 0.402 | 0.355 | 0.387 | 0.339 | 0.379 | 0.380 | 0.400 |
| | 336 | 0.350 | 0.396 | 0.351 | 0.396 | 0.350 | 0.395 | **0.325** | **0.386** | 0.332 | 0.396 | 0.365 | 0.401 | **0.329** | **0.384** | 0.428 | 0.432 |
| | 720 | 0.385 | 0.429 | 0.384 | 0.427 | 0.380 | 0.425 | **0.372** | **0.421** | **0.342** | **0.408** | 0.413 | 0.436 | 0.379 | 0.422 | 0.427 | 0.445 |
| | avg | 0.333 | 0.385 | 0.332 | 0.383 | 0.331 | 0.382 | **0.320** | **0.378** | **0.316** | 0.384 | 0.354 | 0.391 | 0.330 | **0.381** | 0.383 | 0.407 |
| **ETTm1** | 96 | 0.303 | 0.347 | 0.304 | 0.348 | 0.303 | 0.347 | 0.301 | **0.342** | **0.291** | **0.340** | 0.321 | 0.361 | **0.290** | **0.342** | 0.334 | 0.368 |
| | 192 | 0.335 | 0.367 | 0.337 | 0.369 | 0.335 | 0.367 | 0.335 | **0.363** | **0.327** | **0.365** | 0.360 | 0.380 | **0.332** | 0.369 | 0.377 | 0.391 |
| | 336 | 0.367 | 0.386 | 0.368 | 0.387 | **0.366** | 0.385 | 0.370 | **0.383** | **0.360** | **0.381** | 0.390 | 0.404 | **0.366** | 0.392 | 0.426 | 0.420 |
| | 720 | 0.419 | **0.415** | 0.419 | 0.417 | 0.419 | 0.416 | 0.425 | **0.414** | **0.415** | 0.417 | 0.454 | 0.438 | **0.416** | 0.420 | 0.491 | 0.459 |
| | avg | 0.356 | **0.379** | 0.357 | 0.380 | 0.356 | **0.379** | 0.358 | **0.376** | **0.348** | **0.376** | 0.381 | 0.396 | **0.351** | 0.381 | 0.407 | 0.410 |
| **ETTm2** | 96 | **0.163** | **0.254** | **0.162** | **0.253** | **0.163** | **0.253** | 0.164 | **0.253** | 0.164 | **0.254** | 0.173 | 0.257 | 0.165 | 0.255 | 0.180 | 0.264 |
| | 192 | **0.218** | **0.291** | **0.218** | **0.291** | **0.217** | **0.290** | 0.219 | **0.290** | 0.223 | 0.295 | 0.238 | 0.299 | 0.220 | 0.292 | 0.250 | 0.309 |
| | 336 | 0.273 | 0.329 | **0.271** | **0.326** | **0.270** | **0.325** | 0.273 | **0.326** | 0.279 | 0.330 | 0.296 | 0.338 | 0.274 | 0.329 | 0.311 | 0.348 |
| | 720 | **0.361** | **0.383** | **0.359** | **0.381** | **0.359** | **0.381** | 0.366 | 0.385 | **0.359** | **0.383** | 0.393 | 0.395 | 0.362 | 0.385 | 0.412 | 0.407 |
| | avg | 0.254 | 0.314 | **0.253** | **0.313** | **0.252** | **0.312** | 0.256 | 0.314 | 0.256 | 0.316 | 0.275 | 0.322 | 0.255 | 0.315 | 0.288 | 0.332 |
| **Electricity** | 96 | 0.144 | 0.248 | 0.143 | 0.246 | 0.143 | 0.248 | **0.140** | 0.235 | **0.129** | **0.224** | 0.141 | 0.235 | **0.129** | **0.222** | 0.148 | 0.240 |
| | 192 | 0.158 | 0.261 | 0.158 | 0.261 | 0.157 | 0.260 | 0.154 | 0.248 | **0.140** | **0.220** | 0.151 | 0.247 | **0.147** | **0.240** | 0.162 | 0.253 |
| | 336 | 0.175 | 0.276 | 0.176 | 0.277 | 0.174 | 0.275 | 0.171 | 0.264 | **0.161** | **0.255** | 0.173 | 0.267 | **0.163** | **0.259** | 0.178 | 0.269 |
| | 720 | 0.214 | 0.308 | 0.214 | 0.308 | 0.213 | 0.307 | 0.209 | 0.297 | **0.194** | **0.287** | 0.201 | 0.293 | **0.197** | **0.290** | 0.225 | 0.317 |
| | avg | 0.172 | 0.273 | 0.173 | 0.273 | 0.172 | 0.272 | 0.169 | 0.261 | **0.156** | **0.247** | 0.167 | 0.261 | **0.159** | **0.253** | 0.178 | 0.270 |
| **Weather** | 96 | 0.172 | 0.226 | 0.172 | 0.227 | 0.173 | 0.227 | 0.175 | 0.225 | **0.147** | **0.197** | 0.162 | 0.207 | **0.149** | **0.198** | 0.174 | 0.214 |
| | 192 | 0.216 | 0.262 | 0.216 | 0.262 | 0.216 | 0.263 | 0.218 | 0.260 | **0.189** | **0.239** | 0.208 | 0.248 | **0.194** | **0.241** | 0.221 | 0.254 |
| | 336 | 0.261 | 0.295 | 0.261 | 0.295 | 0.262 | 0.296 | 0.265 | 0.294 | **0.241** | **0.280** | 0.263 | 0.290 | **0.245** | **0.282** | 0.278 | 0.296 |
| | 720 | 0.326 | 0.341 | 0.326 | 0.342 | 0.326 | 0.342 | 0.329 | 0.339 | **0.310** | **0.330** | 0.340 | 0.341 | **0.314** | **0.334** | 0.358 | 0.347 |
| | avg | 0.244 | 0.281 | 0.244 | 0.281 | 0.244 | 0.282 | 0.247 | 0.280 | **0.222** | **0.262** | 0.243 | 0.272 | **0.226** | **0.264** | 0.258 | 0.278 |

*Table 14.* Experiment 1 (Dissecting the Backbone): Marginalized performance and stability with 95% Confidence Intervals.

*(a)* Experiment 1 (Dissecting the Backbone): Performance evaluation and statistical reliability analysis: Comparison of five configurations with 95% Confidence Intervals

| Dataset | H | Point wise | | | Patch wise | | | Variate wise | | | Identity | | | L to dim | | |
|---|---|---|---|---|---|---|---|---|---|---|---|---|---|---|---|---|
| | | $\hat{\mu}$ | $ci_{low}$ | $ci_{upp}$ | $\hat{\mu}$ | $ci_{low}$ | $ci_{upp}$ | $\hat{\mu}$ | $ci_{low}$ | $ci_{upp}$ | $\hat{\mu}$ | $ci_{low}$ | $ci_{upp}$ | $\hat{\mu}$ | $ci_{low}$ | $ci_{upp}$ |
| **ETTh1** | 96 | 0.3898 | 0.3865 | 0.3932 | **0.3772** | 0.3758 | 0.3786 | **0.3834** | 0.3816 | 0.3853 | 0.5890 | 0.5530 | 0.6250 | 0.3937 | 0.3900 | 0.3974 |
| | 192 | 0.4324 | 0.4286 | 0.4363 | **0.4240** | 0.4208 | 0.4271 | **0.4247** | 0.4217 | 0.4277 | 0.6092 | 0.5760 | 0.6424 | 0.4343 | 0.4305 | 0.4381 |
| | 336 | 0.4651 | 0.4606 | 0.4696 | **0.4575** | 0.4536 | 0.4614 | **0.4589** | 0.4555 | 0.4624 | 0.6262 | 0.5983 | 0.6541 | 0.4693 | 0.4656 | 0.4730 |
| | 720 | 0.4772 | 0.4663 | 0.4881 | 0.4812 | 0.4727 | 0.4898 | **0.4618** | 0.4581 | 0.4654 | 0.6317 | 0.5975 | 0.6659 | **0.4751** | 0.4692 | 0.4810 |
| | avg | 0.4394 | 0.4345 | 0.4444 | **0.4327** | 0.4276 | 0.4379 | **0.4308** | 0.4267 | 0.4349 | 0.6132 | 0.5967 | 0.6297 | 0.4417 | 0.4373 | 0.4461 |
| **ETTh2** | 96 | 0.3076 | 0.3007 | 0.3145 | **0.2912** | 0.2884 | 0.2940 | **0.2963** | 0.2921 | 0.3006 | 0.3438 | 0.3321 | 0.3556 | 0.3034 | 0.2989 | 0.3080 |
| | 192 | 0.3799 | 0.3740 | 0.3858 | **0.3626** | 0.3592 | 0.3660 | **0.3709** | 0.3667 | 0.3751 | 0.3969 | 0.3903 | 0.4034 | 0.3752 | 0.3711 | 0.3794 |
| | 336 | 0.4123 | 0.4057 | 0.4189 | **0.3949** | 0.3903 | 0.3996 | **0.3937** | 0.3892 | 0.3981 | 0.4102 | 0.4038 | 0.4166 | 0.3998 | 0.3951 | 0.4046 |
| | 720 | 0.4365 | 0.4275 | 0.4456 | 0.4205 | 0.4161 | 0.4249 | **0.4137** | 0.4102 | 0.4172 | 0.4408 | 0.4320 | 0.4495 | **0.4203** | 0.4160 | 0.4247 |
| | avg | 0.3866 | 0.3801 | 0.3931 | **0.3698** | 0.3640 | 0.3756 | **0.3713** | 0.3659 | 0.3767 | 0.3997 | 0.3941 | 0.4054 | 0.3774 | 0.3719 | 0.3829 |
| **ETTm1** | 96 | 0.3134 | 0.3103 | 0.3164 | **0.3054** | 0.3023 | 0.3085 | **0.3117** | 0.3085 | 0.3149 | 0.5506 | 0.5119 | 0.5893 | 0.3212 | 0.3174 | 0.3250 |
| | 192 | 0.3558 | 0.3515 | 0.3602 | **0.3483** | 0.3436 | 0.3530 | **0.3515** | 0.3468 | 0.3561 | 0.5708 | 0.5303 | 0.6113 | 0.3596 | 0.3547 | 0.3644 |
| | 336 | 0.3889 | 0.3844 | 0.3935 | **0.3776** | 0.3738 | 0.3814 | **0.3866** | 0.3811 | 0.3921 | 0.5908 | 0.5509 | 0.6307 | 0.3923 | 0.3875 | 0.3972 |
| | 720 | **0.4391** | 0.4357 | 0.4424 | **0.4309** | 0.4280 | 0.4338 | 0.4393 | 0.4356 | 0.4430 | 0.6190 | 0.5876 | 0.6503 | 0.4469 | 0.4428 | 0.4511 |
| | avg | 0.3730 | 0.3669 | 0.3791 | **0.3641** | 0.3581 | 0.3702 | **0.3711** | 0.3648 | 0.3773 | 0.5821 | 0.5631 | 0.6011 | 0.3786 | 0.3724 | 0.3849 |
| **ETTm2** | 96 | 0.1800 | 0.1781 | 0.1820 | **0.1728** | 0.1717 | 0.1740 | **0.1758** | 0.1742 | 0.1773 | 0.2235 | 0.2145 | 0.2326 | 0.1779 | 0.1763 | 0.1795 |
| | 192 | 0.2414 | 0.2377 | 0.2452 | **0.2331** | 0.2309 | 0.2352 | **0.2343** | 0.2314 | 0.2372 | 0.2739 | 0.2645 | 0.2834 | 0.2370 | 0.2337 | 0.2402 |
| | 336 | 0.3019 | 0.2977 | 0.3060 | **0.2911** | 0.2886 | 0.2936 | **0.2928** | 0.2896 | 0.2961 | 0.3218 | 0.3154 | 0.3281 | 0.2968 | 0.2936 | 0.3000 |
| | 720 | 0.3981 | 0.3930 | 0.4032 | **0.3813** | 0.3780 | 0.3846 | **0.3863** | 0.3823 | 0.3903 | 0.4095 | 0.4040 | 0.4151 | 0.3892 | 0.3851 | 0.3933 |
| | avg | 0.2782 | 0.2680 | 0.2885 | **0.2680** | 0.2583 | 0.2776 | **0.2708** | 0.2610 | 0.2806 | 0.3056 | 0.2962 | 0.3150 | 0.2735 | 0.2636 | 0.2835 |
| **Electricity** | 96 | 0.1609 | 0.1516 | 0.1702 | **0.1512** | 0.1464 | 0.1560 | **0.1527** | 0.1489 | 0.1564 | 0.1639 | 0.1545 | 0.1734 | 0.1573 | 0.1521 | 0.1626 |
| | 192 | 0.1716 | 0.1671 | 0.1762 | **0.1641** | 0.1618 | 0.1665 | **0.1710** | 0.1687 | 0.1733 | 0.1777 | 0.1716 | 0.1837 | **0.1710** | 0.1685 | 0.1734 |
| | 336 | 0.1917 | 0.1862 | 0.1971 | **0.1828** | 0.1797 | 0.1860 | **0.1906** | 0.1871 | 0.1942 | 0.2270 | 0.1800 | 0.2740 | 0.1928 | 0.1888 | 0.1967 |
| | 720 | 0.2342 | 0.2271 | 0.2412 | **0.2236** | 0.2195 | 0.2277 | 0.2293 | 0.2240 | 0.2345 | 0.2335 | 0.2250 | 0.2420 | **0.2218** | 0.2180 | 0.2255 |
| | avg | 0.1890 | 0.1836 | 0.1944 | **0.1798** | 0.1763 | 0.1833 | 0.1856 | 0.1820 | 0.1893 | 0.2019 | 0.1866 | 0.2172 | **0.1849** | 0.1816 | 0.1882 |
| **Weather** | 96 | 0.1715 | 0.1689 | 0.1741 | **0.1673** | 0.1644 | 0.1703 | **0.1679** | 0.1652 | 0.1707 | 0.2120 | 0.2054 | 0.2186 | 0.1705 | 0.1681 | 0.1728 |
| | 192 | 0.2182 | 0.2152 | 0.2213 | **0.2142** | 0.2111 | 0.2173 | **0.2144** | 0.2113 | 0.2176 | 0.2519 | 0.2466 | 0.2572 | 0.2147 | 0.2122 | 0.2171 |
| | 336 | 0.2639 | 0.2613 | 0.2665 | **0.2596** | 0.2568 | 0.2625 | **0.2593** | 0.2568 | 0.2618 | 0.2950 | 0.2894 | 0.3007 | 0.2610 | 0.2585 | 0.2635 |
| | 720 | 0.3391 | 0.3357 | 0.3425 | **0.3357** | 0.3322 | 0.3393 | **0.3345** | 0.3309 | 0.3381 | 0.3594 | 0.3545 | 0.3642 | 0.3386 | 0.3353 | 0.3419 |
| | avg | 0.2391 | 0.2315 | 0.2466 | **0.2363** | 0.2289 | 0.2437 | **0.2362** | 0.2289 | 0.2435 | 0.2721 | 0.2645 | 0.2796 | 0.2384 | 0.2309 | 0.2459 |

*Table 15.* Experiment 1 (Dissecting the Backbone): Performance of 6 Datasets across 4 Horizons on 3 Encoders with 95% Confidence Interval

| Dataset | H | MLP | | | | | Transformer | | | | | Identity | | | | |
|---|---|---|---|---|---|---|---|---|---|---|---|---|---|---|---|---|
| | | $\hat{\mu}$ | $ci_{low}$ | $ci_{upp}$ | $\hat{\sigma}$ | $min$ | $\hat{\mu}$ | $ci_{low}$ | $ci_{upp}$ | $\hat{\sigma}$ | $min$ | $\hat{\mu}$ | $ci_{low}$ | $ci_{upp}$ | $\hat{\sigma}$ | $min$ |
| ETTh1 | 96 | 0.4267 | 0.4130 | 0.4404 | 0.0816 | 0.3700 | 0.4406 | 0.4255 | 0.4557 | 0.0930 | 0.3721 | 0.3837 | 0.3810 | 0.3863 | 0.0231 | 0.3641 |
| | 192 | 0.4745 | 0.4610 | 0.4879 | 0.0827 | 0.4043 | 0.4852 | 0.4709 | 0.4994 | 0.0988 | 0.4091 | 0.4242 | 0.4206 | 0.4278 | 0.0306 | 0.3983 |
| | 336 | 0.5002 | 0.4892 | 0.5112 | 0.0787 | 0.4335 | 0.5069 | 0.4957 | 0.5181 | 0.0795 | 0.4406 | 0.4623 | 0.4585 | 0.4661 | 0.0268 | 0.4196 |
| | 720 | 0.5251 | 0.5103 | 0.5399 | 0.1039 | 0.4533 | 0.5412 | 0.5259 | 0.5565 | 0.1014 | 0.4504 | 0.4600 | 0.4552 | 0.4648 | 0.0406 | 0.4248 |
| | avg | 0.4798 | 0.4726 | 0.4870 | 0.0938 | - | 0.4911 | 0.4835 | 0.4987 | 0.1004 | - | 0.4316 | 0.4283 | 0.4349 | 0.0424 | - |
| ETTh2 | 96 | 0.3276 | 0.3210 | 0.3342 | 0.0423 | 0.2770 | 0.3370 | 0.3312 | 0.3427 | 0.0437 | 0.2835 | 0.2890 | 0.2849 | 0.2931 | 0.0368 | 0.2656 |
| | 192 | 0.3916 | 0.3876 | 0.3955 | 0.0293 | 0.3468 | 0.3978 | 0.3938 | 0.4017 | 0.0310 | 0.3485 | 0.3645 | 0.3605 | 0.3686 | 0.0373 | 0.3258 |
| | 336 | 0.4117 | 0.4079 | 0.4155 | 0.0288 | 0.3680 | 0.4224 | 0.4189 | 0.4260 | 0.0308 | 0.3788 | 0.3914 | 0.3871 | 0.3957 | 0.0325 | 0.3502 |
| | 720 | 0.4356 | 0.4320 | 0.4393 | 0.0339 | 0.4034 | 0.4517 | 0.4465 | 0.4569 | 0.0378 | 0.4065 | 0.4109 | 0.4068 | 0.4149 | 0.0374 | 0.3799 |
| | avg | 0.3937 | 0.3898 | 0.3976 | 0.0473 | - | 0.4042 | 0.4002 | 0.4083 | 0.0512 | - | 0.3666 | 0.3624 | 0.3709 | 0.0547 | - |
| ETTm1 | 96 | 0.3446 | 0.3295 | 0.3597 | 0.0785 | 0.2837 | 0.3607 | 0.3450 | 0.3764 | 0.0956 | 0.2866 | 0.3191 | 0.3160 | 0.3221 | 0.0185 | 0.3026 |
| | 192 | 0.3866 | 0.3710 | 0.4022 | 0.0695 | 0.3235 | 0.3984 | 0.3822 | 0.4146 | 0.0890 | 0.3298 | 0.3592 | 0.3550 | 0.3633 | 0.0221 | 0.3354 |
| | 336 | 0.4187 | 0.4034 | 0.4340 | 0.0734 | 0.3581 | 0.4284 | 0.4126 | 0.4441 | 0.0876 | 0.3650 | 0.3922 | 0.3881 | 0.3963 | 0.0199 | 0.3658 |
| | 720 | 0.4701 | 0.4586 | 0.4815 | 0.0697 | 0.4128 | 0.4759 | 0.4640 | 0.4878 | 0.0830 | 0.4129 | 0.4397 | 0.4369 | 0.4425 | 0.0158 | 0.4185 |
| | avg | 0.4035 | 0.3952 | 0.4118 | 0.0912 | - | 0.4148 | 0.4064 | 0.4231 | 0.0997 | - | 0.3762 | 0.3719 | 0.3805 | 0.0457 | - |
| ETTm2 | 96 | 0.1879 | 0.1846 | 0.1912 | 0.0197 | 0.1667 | 0.1975 | 0.1939 | 0.2011 | 0.0265 | 0.1652 | 0.1732 | 0.1721 | 0.1744 | 0.0087 | 0.1622 |
| | 192 | 0.2492 | 0.2451 | 0.2532 | 0.0213 | 0.2201 | 0.2559 | 0.2524 | 0.2594 | 0.0221 | 0.2313 | 0.2308 | 0.2287 | 0.2330 | 0.0133 | 0.2170 |
| | 336 | 0.3067 | 0.3036 | 0.3099 | 0.0198 | 0.2738 | 0.3154 | 0.3125 | 0.3182 | 0.0199 | 0.2794 | 0.2901 | 0.2875 | 0.2926 | 0.0194 | 0.2701 |
| | 720 | 0.4009 | 0.3974 | 0.4044 | 0.0232 | 0.3634 | 0.4094 | 0.4060 | 0.4129 | 0.0265 | 0.3653 | 0.3837 | 0.3804 | 0.3871 | 0.0270 | 0.3591 |
| | avg | 0.2846 | 0.2774 | 0.2917 | 0.0759 | - | 0.2929 | 0.2858 | 0.3000 | 0.0772 | - | 0.2679 | 0.2609 | 0.2749 | 0.0756 | - |
| Electricity | 96 | 0.1580 | 0.1537 | 0.1622 | 0.0171 | 0.1338 | 0.1498 | 0.1452 | 0.1545 | 0.0194 | 0.1315 | 0.1645 | 0.1598 | 0.1693 | 0.0254 | 0.1428 |
| | 192 | 0.1713 | 0.1690 | 0.1735 | 0.0130 | 0.1515 | 0.1644 | 0.1623 | 0.1665 | 0.0129 | 0.1465 | 0.1731 | 0.1706 | 0.1756 | 0.0161 | 0.1574 |
| | 336 | 0.1918 | 0.1886 | 0.1951 | 0.0158 | 0.1681 | 0.1851 | 0.1817 | 0.1885 | 0.0153 | 0.1644 | 0.1928 | 0.1897 | 0.1959 | 0.0179 | 0.1735 |
| | 720 | 0.2282 | 0.2242 | 0.2322 | 0.0171 | 0.2070 | 0.2196 | 0.2153 | 0.2239 | 0.0171 | 0.1972 | 0.2326 | 0.2288 | 0.2364 | 0.0191 | 0.2132 |
| | avg | 0.1867 | 0.1836 | 0.1899 | 0.0253 | - | 0.1789 | 0.1756 | 0.1823 | 0.0242 | avg | 0.1895 | 0.1866 | 0.1924 | 0.0275 | - |
| Weather | 96 | 0.1713 | 0.1681 | 0.1744 | 0.0180 | 0.1468 | 0.1731 | 0.1702 | 0.1760 | 0.0198 | 0.1474 | 0.1780 | 0.1766 | 0.1795 | 0.0101 | 0.1587 |
| | 192 | 0.2179 | 0.2145 | 0.2213 | 0.0192 | 0.1914 | 0.2248 | 0.2208 | 0.2288 | 0.0235 | 0.1912 | 0.2245 | 0.2230 | 0.2260 | 0.0087 | 0.2090 |
| | 336 | 0.2673 | 0.2639 | 0.2707 | 0.0224 | 0.2415 | 0.2663 | 0.2631 | 0.2696 | 0.0237 | 0.2422 | 0.2707 | 0.2693 | 0.2722 | 0.0084 | 0.2605 |
| | 720 | 0.3435 | 0.3400 | 0.3470 | 0.0192 | 0.3131 | 0.3457 | 0.3421 | 0.3492 | 0.0209 | 0.3179 | 0.3424 | 0.3399 | 0.3448 | 0.0129 | 0.3257 |
| | avg | 0.2415 | 0.2359 | 0.2470 | 0.0569 | - | 0.2439 | 0.2382 | 0.2497 | 0.0588 | - | 0.2465 | 0.2414 | 0.2516 | 0.0514 | - |

*Table 16.* Experiment 2 (Auditing Input Transformations): Marginalized performance and stability with 95% Confidence Intervals. Highlighting follows the convention in Table 2.

(a) Performance across 2 Embeddings.

| Dataset | Channel | | | | | Point | | | | |
|---|---|---|---|---|---|---|---|---|---|---|
| | $\hat{\mu}$ | $ci_{low}$ | $ci_{upp}$ | $\hat{\sigma}$ | $B$ | $\hat{\mu}$ | $ci_{low}$ | $ci_{upp}$ | $\hat{\sigma}$ | $B$ |
| **ETTh1** | 0.4028 | 0.3943 | 0.4114 | 0.0014 | **0.3743** | **0.3866** | 0.3845 | 0.3887 | 0.0001 | 0.3747 |
| **ETTh2** | **0.3010** | 0.2987 | 0.3033 | 0.0002 | **0.2818** | 0.3028 | 0.2987 | 0.3069 | 0.0006 | 0.2829 |
| **ETTm1** | 0.3462 | 0.3431 | 0.3492 | 0.0003 | 0.3160 | **0.3362** | 0.3334 | 0.3390 | 0.0003 | **0.3137** |
| **ETTm2** | 0.1830 | 0.1822 | 0.1838 | 0.0000 | **0.1738** | **0.1822** | 0.1818 | 0.1826 | 0.0000 | 0.1778 |
| **Electricity** | 0.2018 | 0.1971 | 0.2066 | 0.0008 | **0.1580** | **0.1976** | 0.1959 | 0.1993 | 0.0001 | 0.1840 |
| **Weather** | **0.1903** | 0.1881 | 0.1925 | 0.0001 | **0.1599** | 0.1913 | 0.1904 | 0.1922 | 0.0000 | 0.1816 |

(b) Performance across different Encoders.

| Dataset | Identity | | | | | MLP | | | | | Transformer | | | | |
|---|---|---|---|---|---|---|---|---|---|---|---|---|---|---|---|
| | $\hat{\mu}$ | $ci_{low}$ | $ci_{upp}$ | $\hat{\sigma}$ | $min$ | $\hat{\mu}$ | $ci_{low}$ | $ci_{upp}$ | $\hat{\sigma}$ | $min$ | $\hat{\mu}$ | $ci_{low}$ | $ci_{upp}$ | $\hat{\sigma}$ | $min$ |
| **ETTh1** | **0.3932** | 0.3881 | 0.3982 | 0.0006 | **0.3743** | **0.3971** | 0.3837 | 0.4104 | 0.0011 | **0.3749** | 0.3984 | 0.3852 | 0.4116 | 0.0011 | 0.3773 |
| **ETTh2** | **0.2908** | 0.2890 | 0.2926 | 0.0001 | **0.2829** | **0.3052** | 0.3027 | 0.3077 | 0.0002 | **0.2818** | 0.3097 | 0.3040 | 0.3155 | 0.0008 | 0.2873 |
| **ETTm1** | 0.3613 | 0.3599 | 0.3627 | 0.0001 | 0.3522 | **0.3314** | 0.3290 | 0.3339 | 0.0002 | **0.3159** | **0.3308** | 0.3277 | 0.3338 | 0.0002 | **0.3137** |
| **ETTm2** | **0.1827** | 0.1823 | 0.1830 | 0.0000 | 0.1804 | **0.1805** | 0.1788 | 0.1822 | 0.0000 | **0.1738** | 0.1843 | 0.1830 | 0.1857 | 0.0000 | **0.1770** |
| **Electricity** | 0.2078 | 0.2048 | 0.2109 | 0.0002 | 0.1981 | **0.1930** | 0.1886 | 0.1973 | 0.0005 | **0.1588** | **0.1983** | 0.1933 | 0.2033 | 0.0006 | **0.1580** |
| **Weather** | 0.1935 | 0.1931 | 0.1939 | 0.0000 | 0.1899 | **0.1807** | 0.1761 | 0.1853 | 0.0001 | **0.1599** | **0.1901** | 0.1873 | 0.1929 | 0.0000 | **0.1785** |

(c) Performance across different Input Transformations.

| Dataset | BaseLine | | | | | Cycle | | | | | MultiScale | | | | | TrendSeasonal | | | | |
|---|---|---|---|---|---|---|---|---|---|---|---|---|---|---|---|---|---|---|---|---|
| | $\hat{\mu}$ | $ci_{low}$ | $ci_{upp}$ | $\hat{\sigma}$ | $min$ | $\hat{\mu}$ | $ci_{low}$ | $ci_{upp}$ | $\hat{\sigma}$ | $min$ | $\hat{\mu}$ | $ci_{low}$ | $ci_{upp}$ | $\hat{\sigma}$ | $min$ | $\hat{\mu}$ | $ci_{low}$ | $ci_{upp}$ | $\hat{\sigma}$ | $min$ |
| **ETTh1** | 0.3975 | 0.3902 | 0.4048 | 0.0010 | **0.3749** | **0.3891** | 0.3777 | 0.4004 | 0.0008 | **0.3743** | **0.3921** | 0.3848 | 0.3994 | 0.0003 | 0.3829 | 0.3947 | 0.3848 | 0.4046 | 0.0006 | 0.3811 |
| **ETTh2** | **0.3013** | 0.2986 | 0.3040 | 0.0001 | 0.2868 | **0.2969** | 0.2943 | 0.2996 | 0.0001 | **0.2818** | 0.3071 | 0.2991 | 0.3150 | 0.0012 | 0.2874 | 0.3024 | 0.2993 | 0.3055 | 0.0002 | **0.2849** |
| **ETTm1** | 0.3451 | 0.3404 | 0.3499 | 0.0004 | **0.3139** | 0.3431 | 0.3386 | 0.3475 | 0.0004 | **0.3137** | **0.3380** | 0.3341 | 0.3420 | 0.0003 | 0.3159 | **0.3384** | 0.3346 | 0.3423 | 0.0003 | 0.3166 |
| **ETTm2** | 0.1827 | 0.1818 | 0.1835 | 0.0000 | **0.1738** | **0.1825** | 0.1818 | 0.1833 | 0.0000 | **0.1804** | 0.1830 | 0.1825 | 0.1834 | 0.0000 | 0.1818 | **0.1821** | 0.1815 | 0.1826 | 0.0000 | 0.1805 |
| **Electricity** | 0.2061 | 0.2006 | 0.2116 | 0.0006 | **0.1679** | 0.1999 | 0.1947 | 0.2051 | 0.0005 | **0.1580** | **0.1952** | 0.1905 | 0.1999 | 0.0004 | **0.1580** | **0.1976** | 0.1931 | 0.2022 | 0.0004 | 0.1729 |
| **Weather** | **0.1882** | 0.1860 | 0.1904 | 0.0001 | **0.1599** | **0.1917** | 0.1910 | 0.1924 | 0.0000 | **0.1899** | 0.1950 | 0.1942 | 0.1959 | 0.0000 | 0.1915 | 0.1934 | 0.1931 | 0.1937 | 0.0000 | 0.1920 |

*Table 17.* Experiment 3 (Revisiting Spectral Processing): Performance of 6 Datasets across 4 Horizons on 3 Attentions.

| Dataset | H | Transformer | | | | | GeomAttention | | | | | Variate wise and Identity | | | | |
|---|---|---|---|---|---|---|---|---|---|---|---|---|---|---|---|---|
| | | $\mu$ | $ci_{low}$ | $ci_{upp}$ | $\hat{\sigma}$ | $min$ | $\mu$ | $ci_{low}$ | $ci_{upp}$ | $\hat{\sigma}$ | $min$ | $\mu$ | $ci_{low}$ | $ci_{upp}$ | $\hat{\sigma}$ | $min$ |
| ETTh1 | 96 | 0.3934 | 0.3882 | 0.3985 | 0.0141 | 0.3789 | **0.3812** | 0.3783 | 0.3841 | 0.0100 | **0.3730** | **0.3805** | 0.3761 | 0.3848 | 0.0098 | **0.3641** |
| | 192 | 0.4354 | 0.4316 | 0.4392 | 0.0100 | 0.4171 | **0.4287** | 0.4246 | 0.4327 | 0.0100 | **0.4148** | **0.4199** | 0.4121 | 0.4277 | 0.0179 | **0.3983** |
| | 336 | 0.4656 | 0.4608 | 0.4704 | 0.0141 | 0.4434 | **0.4601** | 0.4552 | 0.4651 | 0.0141 | **0.4314** | **0.4583** | 0.4496 | 0.4671 | 0.0204 | **0.4249** |
| | 720 | 0.4711 | 0.4645 | 0.4778 | 0.0141 | 0.4504 | **0.4646** | 0.4581 | 0.4710 | 0.0141 | **0.4473** | **0.4520** | 0.4424 | 0.4617 | 0.0191 | **0.4366** |
| | avg | 0.4401 | 0.4336 | 0.4467 | 0.0332 | - | **0.4324** | 0.4254 | 0.4393 | 0.0361 | - | **0.4271** | 0.4191 | 0.4351 | 0.0355 | - |
| ETTh2 | 96 | 0.3214 | 0.3117 | 0.3312 | 0.0224 | 0.2964 | **0.2945** | 0.2918 | 0.2973 | 0.0000 | **0.2840** | **0.2809** | 0.2755 | 0.2864 | 0.0100 | **0.2656** |
| | 192 | 0.3887 | 0.3826 | 0.3948 | 0.0173 | 0.3599 | **0.3709** | 0.3669 | 0.3749 | 0.0100 | **0.3520** | **0.3584** | 0.3493 | 0.3674 | 0.0200 | **0.3264** |
| | 336 | 0.4076 | 0.3997 | 0.4155 | 0.0200 | 0.3809 | **0.4030** | 0.3953 | 0.4107 | 0.0200 | **0.3727** | **0.3852** | 0.3736 | 0.3967 | 0.0265 | **0.3509** |
| | 720 | 0.4309 | 0.4216 | 0.4401 | 0.0224 | 0.4065 | **0.4148** | 0.4099 | 0.4197 | 0.0100 | **0.3921** | **0.4030** | 0.3960 | 0.4099 | 0.0141 | **0.3841** |
| | avg | 0.3894 | 0.3809 | 0.3979 | 0.0436 | - | **0.3736** | 0.3646 | 0.3827 | 0.0458 | - | **0.3597** | 0.3486 | 0.3708 | 0.0490 | - |
| ETTm1 | 96 | 0.3507 | 0.3135 | 0.3879 | 0.1020 | **0.2987** | 0.2974 | 0.2915 | 0.3034 | 0.0173 | **0.2825** | **0.3260** | 0.3157 | 0.3364 | 0.0245 | 0.3040 |
| | 192 | 0.3670 | 0.3458 | 0.3883 | 0.0520 | **0.3319** | 0.3454 | 0.3378 | 0.3530 | 0.0173 | **0.3232** | **0.3584** | 0.3457 | 0.3710 | 0.0265 | 0.3367 |
| | 336 | 0.4044 | 0.3784 | 0.4305 | 0.0608 | **0.3655** | 0.3792 | 0.3716 | 0.3869 | 0.0173 | **0.3590** | **0.3932** | 0.3792 | 0.4073 | 0.0265 | 0.3676 |
| | 720 | **0.4447** | 0.4374 | 0.4520 | 0.0200 | 0.4194 | 0.4334 | 0.4269 | 0.4399 | 0.0173 | **0.4183** | 0.4478 | 0.4359 | 0.4598 | 0.0265 | **0.4192** |
| | avg | 0.3911 | 0.3761 | 0.4061 | 0.0762 | - | 0.3624 | 0.3516 | 0.3732 | 0.0548 | - | **0.3795** | 0.3671 | 0.3919 | 0.0548 | - |
| ETTm2 | 96 | 0.1980 | 0.1876 | 0.2084 | 0.0283 | 0.1782 | **0.1738** | 0.1717 | 0.1760 | 0.0000 | **0.1646** | **0.1731** | 0.1697 | 0.1766 | 0.0079 | **0.1622** |
| | 192 | 0.2478 | 0.2453 | 0.2504 | 0.0000 | 0.2399 | **0.2332** | 0.2297 | 0.2367 | 0.0100 | **0.2229** | **0.2296** | 0.2238 | 0.2354 | 0.0119 | **0.2184** |
| | 336 | 0.3080 | 0.3025 | 0.3135 | 0.0141 | 0.2894 | **0.2930** | 0.2878 | 0.2982 | 0.0141 | **0.2735** | **0.2866** | 0.2802 | 0.2930 | 0.0150 | **0.2707** |
| | 720 | 0.3981 | 0.3906 | 0.4056 | 0.0200 | 0.3696 | **0.3846** | 0.3781 | 0.3911 | 0.0173 | **0.3639** | **0.3793** | 0.3716 | 0.3870 | 0.0184 | **0.3594** |
| | avg | 0.2870 | 0.2713 | 0.3027 | 0.0800 | - | **0.2697** | 0.2536 | 0.2859 | 0.0825 | - | **0.2721** | 0.2545 | 0.2897 | 0.0798 | - |
| Electricity | 96 | **0.1517** | 0.1434 | 0.1600 | 0.0173 | **0.1335** | **0.1537** | 0.1449 | 0.1625 | 0.0200 | **0.1322** | 0.1702 | 0.1574 | 0.1830 | 0.0270 | 0.1428 |
| | 192 | **0.1705** | 0.1655 | 0.1755 | 0.0141 | **0.1555** | **0.1701** | 0.1646 | 0.1757 | 0.0173 | **0.1493** | 0.1810 | 0.1735 | 0.1885 | 0.0184 | 0.1584 |
| | 336 | **0.1930** | 0.1850 | 0.2011 | 0.0224 | **0.1722** | **0.1929** | 0.1841 | 0.2017 | 0.0224 | **0.1648** | 0.2015 | 0.1915 | 0.2116 | 0.0230 | 0.1756 |
| | 720 | **0.2287** | 0.2180 | 0.2393 | 0.0245 | **0.1972** | **0.2244** | 0.2127 | 0.2361 | 0.0283 | **0.1964** | 0.2448 | 0.2333 | 0.2564 | 0.0236 | 0.2137 |
| | avg | **0.1858** | 0.1793 | 0.1923 | 0.0332 | - | **0.1851** | 0.1787 | 0.1915 | 0.0332 | - | 0.1974 | 0.1895 | 0.2053 | 0.0351 | - |
| Weather | 96 | **0.1738** | 0.1656 | 0.1821 | 0.0224 | **0.1544** | **0.1564** | 0.1522 | 0.1606 | 0.0100 | **0.1440** | 0.1830 | 0.1799 | 0.1861 | 0.0100 | 0.1724 |
| | 192 | **0.2230** | 0.2140 | 0.2321 | 0.0224 | **0.1976** | **0.2030** | 0.1982 | 0.2079 | 0.0100 | **0.1873** | 0.2267 | 0.2228 | 0.2305 | 0.0100 | 0.2155 |
| | 336 | **0.2610** | 0.2550 | 0.2670 | 0.0141 | **0.2465** | **0.2524** | 0.2472 | 0.2575 | 0.0141 | **0.2376** | 0.2724 | 0.2668 | 0.2780 | 0.0100 | 0.2605 |
| | 720 | **0.3384** | 0.3315 | 0.3453 | 0.0141 | **0.3179** | **0.3305** | 0.3242 | 0.3369 | 0.0141 | **0.3137** | 0.3412 | 0.3339 | 0.3486 | 0.0141 | 0.3259 |
| | avg | **0.2412** | 0.2290 | 0.2534 | 0.0624 | - | **0.2274** | 0.2147 | 0.2400 | 0.0640 | - | 0.2499 | 0.2360 | 0.2638 | 0.0608 | - |

*Table 18.* Mann–Whitney U test p-values (one-tailed, $\alpha = 0.05$) for Identity Encoder vs. other encoders per dataset. Significant results ($p < 0.05$) are shown in **bold**.

| Dataset | vs. Transformer | vs. MLP |
|---|---|---|
| ECL | 0.9903 | 0.8863 |
| ETTh1 | <**0.0001** | <**0.0001** |
| ETTh2 | <**0.0001** | <**0.0001** |
| ETTm1 | 0.4845 | 0.9647 |
| ETTm2 | **0.0041** | 0.1111 |
| Weather | 0.9859 | 0.9995 |

*Table 19.* MSE results on the Exchange-Rate dataset. PatchLinear and iLinear (Identity Encoder) remain competitive with or outperform full architectures on this non-stationary benchmark.

| pred_len | iLinear | PatchLinear | iTransformer | PatchTST | Avg |
|---|---|---|---|---|---|
| 96 | 0.094 | **0.083** | 0.086 | 0.088 | 0.088 |
| 192 | 0.187 | 0.176 | 0.177 | **0.176** | 0.179 |
| 336 | 0.343 | 0.325 | 0.331 | **0.301** | 0.325 |
| 720 | 0.874 | **0.844** | 0.847 | 0.901 | 0.867 |
| Avg | 0.375 | **0.357** | 0.360 | 0.367 | — |

*Table 20.* Optimal hyperparameter configurations on Electricity (robust case). Both models exhibit consistent configurations across all horizons.

| | PatchLinear (Identity) | | | PatchTST (Transformer) | | |
|---|---|---|---|---|---|---|
| pred_len | seq_len | $d_{\mathrm{model}}$ | $\eta$ | seq_len | $d_{\mathrm{model}}$ | $\eta$ |
| 96 | 512 | 512 | 1e-3 | 512 | 128 | 1e-3 |
| 192 | 512 | 512 | 1e-3 | 512 | 128 | 1e-3 |
| 336 | 512 | 512 | 1e-3 | 512 | 128 | 1e-3 |
| 720 | 512 | 512 | 1e-3 | 512 | 128 | 1e-3 |

*Table 21.* Optimal hyperparameter configurations on ETTm1 (brittle case). PatchTST's optimal configuration varies considerably across horizons, while PatchLinear remains stable.

| | PatchLinear (Identity) | | | PatchTST (Transformer) | | |
|---|---|---|---|---|---|---|
| pred_len | seq_len | $d_{\mathrm{model}}$ | $\eta$ | seq_len | $d_{\mathrm{model}}$ | $\eta$ |
| 96 | 512 | 512 | 1e-3 | 512 | 64 | 1e-3 |
| 192 | 512 | 512 | 1e-3 | 512 | 256 | 1e-4 |
| 336 | 512 | 512 | 1e-3 | 512 | 128 | 1e-4 |
| 720 | 512 | 256 | 1e-3 | 512 | 256 | 1e-3 |

*Table 22.* Multi-seed robustness verification on ETTh1. Both $\hat{\mu}$ and $\hat{\sigma}$ remain highly stable across seeds for both models, confirming that our conclusions are not seed-sensitive.

| Model | Metric | seed 333 | seed 2025 | seed 2026 |
|---|---|---|---|---|
| PatchLinear | $\hat{\mu}$ | 0.4258 | 0.4259 | 0.4255 |
| | $\hat{\sigma}$ | 0.0378 | 0.0374 | 0.0372 |
| PatchTST | $\hat{\mu}$ | 0.4484 | 0.4485 | 0.4455 |
| | $\hat{\sigma}$ | 0.0580 | 0.0563 | 0.0509 |

