# OpenReview forum: "CombinationTS: A Modular Framework for Understanding Time-Series Forecasting Models"
_ICML.cc/2026/Conference — ICML 2026 regular_

### Official Review · Reviewer_R3P2 · 2026-02-22

**Soundness:** 3
**Presentation:** 3
**Significance:** 3
**Originality:** 2
**Overall Recommendation:** 4
**Confidence:** 4

**Summary:**

This work proposes CombinationTS, a modular probabilistic evaluation framework for time-series forecasting architectures and addresses the "Attribution Gap" and "Benchmarking Crisis" in Time-Series Forecasting (TSF). CombinationTS, a framework that decomposes TSF models into five orthogonal modules: Input Transformation, Embedding, Encoder, Decoder, and Output Transformation. They introduce a probabilistic evaluation protocol using EC-sampling to measure Marginalized Effectiveness ($\mu$) and Stability ($\sigma^2$) rather than fragile point estimates. This advances the time series research by more robust evaluation techniques.

**Compliance With Llm Reviewing Policy:**

Affirmed.

**Final Justification:**

Rebuttal have resolved some of my issues and I believe the paper makes a valuable contribution to the domain. Therefore, I am increasing my score.

**Key Questions For Authors:**

1. Why is random seed variability excluded from evaluation condition space Ω, given that stability is a central claim?
2. Can you provide confidence intervals or statistical tests supporting your conclusions?
3. How do results change when models are trained to convergence rather than fixed epochs?

**Limitations:**

yes

**Strengths And Weaknesses:**

Strengths:
1. The distinction between the "Fairness Trap" (fixed configs) and "Best Trap" (cherry-picked peaks) is a genuine and underappreciated problem in TSF benchmarking.
2. The Identity Paradox finding is genuinely interesting and has real implications for how the community should interpret architectural claims.
3. The framework is modular and the open-source commitment is valuable for reproducibility.
4. The three-experiment structure cleanly maps to the three research questions, and the spectral stability finding (frequency-domain as variance reducer, not accuracy improver) is a useful point.

Weaknesses:
1. The paper claims to evaluate stability under stochastic evaluation conditions, but explicitly fixes random seed. This contradicts the core claim of measuring stability across evaluation condition space Ω. Since random seed variability is a major contributor to performance variance.
2. Only point estimate setting is considered, but not probabilistic forecasting paradigm.
3. The paper reports averages and variances but does not include Statistical significance tests.
4. The paper fixes training epochs (30) and early stopping patience (3), which may favor simpler models. Complex models often require longer training.
5. The paper lacks theoretical analysis explaining why identity encoders perform well.
6. Figure 1 is not cited.

---

> ### Author Rebuttal · Authors · 2026-03-31
>
> We sincerely thank the reviewer for recognizing the value of our modular framework in addressing the "Attribution Gap" and the significance of the Identity Paradox. We deeply appreciate your highly constructive critiques regarding evaluation stochasticity, statistical testing, and theoretical analysis, which we address in detail below.
>
> ### **Q1/W1: Seed robustness**
> We appreciate the reviewer's attention to seed robustness. In our distributional evaluation framework, the variance introduced by random seeds is effectively marginalized through extensive sampling of the Evaluation Condition (EC) space.
>
> To empirically verify this, we analyzed the MSE statistics of the **PatchTST** model across three different random seeds (333, 2025, and 2026) on the ETTh1 dataset.
>
> ***Table 1**: PatchTST MSE statistics across different random seeds on ETTh1 dataset.*
>
> | Metric | seed 333 | seed 2025 | seed 2026 |
> |-|-|-|-|
> | $\mu$ | 0.448368 | 0.448524 | 0.445542 |
> | $\sigma$ | 0.057974 | 0.056290 | 0.050897 |
>
> As shown above, the results remain highly consistent across different seeds. This confirms that our evaluation is robust to seed-induced stochasticity, providing a stable foundation for reliable and reproducible performance.
>
> ### **Q2/W3: Statistical significance**
>
> Our confidence intervals are reported in `table 15`, `table 16`, and `table 17` in the Appendix of our paper. We appologize for not citing them in the main text. We will fix all citations in the final version. Additionally, we performed Mann-Whitney U tests (one-tailed, α=0.05) to validate that Identity Encoder achieves significantly lower MSE than other encoders.
>
> ***Table 2**: Mann-Whitney U test p-values (one-tailed, α=0.05) for Identity Encoder vs. other encoders per dataset.*
>
> | Dataset | vs Transformer | vs MLP |
> |-|-|-|
> | ECL | 0.9903 | 0.8863 |
> | ETTh1 | **<0.0001** | **<0.0001** |
> | ETTh2 | **<0.0001** | **<0.0001** |
> | ETTm1 | 0.4845 | 0.9647 |
> | ETTm2 | **0.0041** | 0.1111 |
> | Weather | 0.9859 | 0.9995 |
>
> ***Table 3**: Mann-Whitney U test p-values for Identity Encoder vs. other encoders across all datasets combined.*
>
> | Encoder | p-value | Significant |
> |-|-|-|
> | Transformer | **<0.0001** | Yes |
> | MLP | **0.0115** | Yes |
>
> Identity is significantly better than Transformer in 3/6 datasets and MLP in 2/6 datasets (p<0.05). Overall, Identity achieves significantly lower MSE than both encoders across all datasets combined. We will include these results in the final manuscript.
>
>
> ### **Q3/W4: Training Epochs**
> We acknowledge the reviewer's concern regarding the training epochs and convergence of the models. For end-to-end time series forecasting models, the maximum of 30 epochs is sufficient to ensure model convergence. During our experiments, we recorded the stopping epoch when each model triggered early stopping.
>
> ***Table 4**: Early stopping statistics by encoder type.*
>
> | Category | Early Stop Rate | Mean Epoch | Max Epoch |
> |-|-|-|-|
> | MLP | 100.00% | 7.66 | 26 |
> | Transformer | 99.79% | 7.94 | 26 |
> | Identity | 99.51% | 11.67 | 28 |
>
> These results confirm that nearly all models (99.76%) converged well before the 30-epoch limit via early stopping.
>
> ### **W2: Probabilistic Forecasting**
> We thank the reviewer for pointing this out. We would like to clarify that our proposed evaluation framework is highly extensible and naturally supports probabilistic forecasting paradigms. However, to maintain a focused and manageable scope for our extensive empirical analysis, we restricted our experiments in this paper exclusively to point forecasting settings.
>
> ### **W5: Theoretical Analysis**
>
> We thank the reviewer for this insightful question. We provide a theoretical explanation from a bias-variance perspective combined with token interaction decomposition.
>
> We consider: $\mathbb{E}[(\hat{y} - y)^2] = \mathrm{Bias}^2 + \mathrm{Variance} + \mathrm{Noise}$. Let $Z = E(X) = (z_1, \dots, z_K)$. The target function $g^*(Z)$ can be decomposed as:
>
> $$g^*(Z) = \sum_i g_i(z_i) + \sum_{i<j} g_{ij}(z_i, z_j) + \dots$$
>
> where the first term corresponds to first-order (token-wise) effects, and higher-order terms  capture cross-token interactions. In our framework, embedding $E$ models first-order effects, while encoder $\Phi$ captures higher-order interactions.
>
> When the data is dominated by first-order structure (e.g., trend, shifting, or local patterns), the higher-order terms are negligible, and $E(x)$ already captures most predictive information. In this regime, adding a flexible encoder does not significantly reduce bias but increases variance due to higher model complexity, leading to worse generalization. In contrast, when higher-order interactions are strong (e.g., correlation or nonlinear dependencies), the encoder becomes necessary to reduce bias.
>
> ### **W6: Formatting Issues**
> We thank the reviewer for pointing out these formatting issues. We will fix all formatting issues in the final manuscript.

---

> > ### Author Rebuttal · Reviewer_R3P2 · 2026-04-03
> >
> > I do not fully agree with the authors’ statement that reporting seed robustness only for PatchTST is sufficient to address my question. It is quite possible that the performance of other models also varies with different random seeds.
> >
> > Additionally, based on my experience in this field, training for only 30 epochs is not sufficient for all models to properly converge. Many other factors plays a major role like learning rate and model complexity.
> >
> > As future work, I would recommend that the authors consider evaluating probabilistic forecasting and diffusion-based time-series models.
> >
> > That said, I appreciate the effort put into this work and believe the paper makes a valuable contribution to the domain. Therefore, I am increasing my score.

---

> > > ### Author Response · Authors · 2026-04-03
> > >
> > > Thank you again for your constructive feedback and for increasing your score. We sincerely appreciate your recognition of our effort and contribution.
> > >
> > > Regarding seed robustness, we would like to clarify that we have in fact conducted additional seed robustness experiments for **PatchLinear**. We have included the corresponding results in our response to reviewer `gxzB`, and we hope these additional data can at least partially address your concern that performance variations under different random seeds may also exist for models beyond **PatchTST**.
> > >
> > > We also agree that learning rate settings and model complexity can affect convergence speed. However, as shown in the additional results we provided, for point forecasting time-series models, most models converge within 10 epochs, and more than 99% of the models successfully converge within 30 epochs, even when convergence is slowed by learning rate choices or model complexity. Moreover, with the early stopping patience fixed at 3, increasing the maximum number of training epochs does not change the conclusions of our experiments.
> > >
> > > We truly appreciate your comments regarding the number of training epochs. We will add a more detailed discussion of this issue in the camera-ready version to make this aspect clearer.
> > >
> > > Finally, thank you very much for your suggestion to include probabilistic forecasting methods in the evaluation. We agree that this is an important direction, and we will include it as part of our future work.
> > >
> > > Thank you again for your valuable and constructive comments.

---

### Official Review · Reviewer_c29R · 2026-03-02

**Soundness:** 3
**Presentation:** 1
**Significance:** 4
**Originality:** 3
**Overall Recommendation:** 4
**Confidence:** 3

**Summary:**

The authors present an experimental framework to determine which components of time series forecasting models have the most impact on the models performance. The authors shows that the most important choice is the choice of embeddings, with simple (often parameter-free) alternative showing great results for the encoding and decoding components.

The experiments are done on 6 different popular time series forecasting benchmark datasets, and care is given to reduce the risk that the results presented are dominated by noise. For example, results showing the usefulness of a given module for a component is averaged over all other design variables.

**Compliance With Llm Reviewing Policy:**

Affirmed.

**Final Justification:**

My score for this paper went to a "Weak Accept" mostly due to its significance. While I would be happy for it to be accepted, I am not raising my score to an "Accept" following the authors rebuttal to a lack of clarity in how the results obtained by the method presented in the paper can transfer from one dataset to another, or even from one architecture to another.

**Key Questions For Authors:**

The questions are roughly sorted from most to least important.

* *Question 1*: Are model design decisions based on your results transferable to other datasets? In particular, would creating a model with Patch Wise embedding, Identity Encoder, and Identity Decoder would perform quite well for another dataset?
* *Question 2*: For component choices which shows a large variance, can you show whether the EC for these choices can be reliably tuned, or whether their performance is very chaotic? I would consider a model to be tunable reliably if a small number of EC dimensions explains the majority of the variance, especially if the "best" choice in those dimensions is the same for all datasets. Knowing the quality of components that have been roughly tuned with such a criterion could serve as a middle point between the global average (which disfavours components which are sensitive to certain EC dimensions) and the best result (which favours components which are highly unreliable due to noise or chaotic interactions between EC dimensions).
* *Question 3*: In Table 3, why are the results for TimeMixer incomplete?

**Limitations:**

Yes

**Strengths And Weaknesses:**

The order in each category is roughly from the most to the least important.

**Soundness**
* *Strength 1*: The method of decomposing the impact of each choice of each component by integrating (using a Monte Carlo method) over the other choice is quite reasonable.
* *Strength 2*: A wide variety of component choices are used in the study, including some trivial ones which effectively bypass the need of a component.
* *Weakness 1*: Only using 6 datasets for the experiments is limited, especially with 4 of them coming from the same source. It is not impossible that the conclusions presented in the paper comes from specific properties of these datasets, and could not transfer to datasets without said properties.
* *Weakness 2*: Not having included random noise in the experiments (random initialization, random ordering of the training samples, dropout randomness) prevents the paper from being to distinguish how much of the variance comes controllable sources (other components + hyperparameters) versus uncontrollable sources (said noise, but also chaotic effects). Knowing if a choice of a component is simply unreliable or highly dependent on proper tuning is important.

**Presentation**
* *Weakness 3*: The main text should be as self contained as possible. Section 4.2.2 requiring a reader to check Table 9 (in the appendix) is a major break.
* *Weakness 4*: The result tables should use the standard deviation instead of the variance. As it is, it is hard to determine how big the gap between modules are compared to how variable each module is.
* *Weakness 5*: The Red and Blue bolding + underlining are not correctly explained in the relevant table captions. Also, having both of them tend to overload the tables and make the desired conclusions harder to see.

**Significance**
* *Strength 3*: The problem of properly understanding the relative contributions of various implementation choices in a machine learning model is a widespread one in the whole field. While this paper does not introduce the research targeted at solving this problem (the authors cite the Evaluatology book), bringing more light to the issue is important.
* *Strength 4*: Hopefully, this paper will lead to researchers in the time series forecasting field to be more wary of random models getting good benchmark results.

**Originality**
* *Mixed 1*: The authors cited that their work is grounded in the principles of the Evaluatology book. As I have not read said book, I cannot tell whether the methods used are standard Evaluatology method, simply applied to another field (Time Series Forecasting), or whether they are the authors contributions. Nevertheless, I don't see the originality in this paper being the introduction of this method, but in actually doing the work of applying it to a large enough design space to give trustable results.

---

> ### Author Rebuttal · Authors · 2026-03-31
>
> We sincerely thank the reviewer for recognizing the high significance of our work and our Monte Carlo-based decomposition method. We deeply value your constructive critiques regarding the presentation (table formatting, appendix references) and fully commit to fixing all these issues in the revision. Detailed responses follow.
>
> ### **Q1/W1: Identity Encoder Generalizability**
>
> We appreciate the reviewer's valuable comment regarding the generalizability of the Identity Encoder. In fact, exploring this exact limitation is of great interest to us. As shown in `Table 13` of our Appendix, the Identity Encoder is not always the best choice across all datasets.
>
> To provide more concrete evidence, we conducted an additional, targeted hyperparameter search on the **Exchange-Rate** dataset for two models: **PatchLinear** (Patch-wise Embedding + Identity Encoder + Linear Decoder) and **iLinear** (Variate-wise Embedding + Identity Encoder + Linear Decoder). The results of this exploration are shown below:
>
> ***Table 1**: MSE results on Exchange-Rate dataset for PatchLinear and iLinear across different prediction lengths.*
> | pred_len | iLinear | PatchLinear | iTransformer | PatchTST |
> |----------|---------|-------------|--------------|----------|
> | 96 | 0.094 | **0.083** | 0.086 | 0.088 |
> | 192 | 0.187 | **0.176** | 0.177 | **0.176** |
> | 336 | 0.343 | 0.325 | 0.331 | **0.301** |
> | 720 | 0.874 | **0.844** | 0.847 | 0.901 |
> | **Avg** | 0.375 | **0.357** | 0.360 | 0.367 |
>
> ### **Q2/W2: Variance Analysis of Component Choices**
> We sincerely thank the reviewer for this insightful point. We fully agree that a component should not be deemed ineffective if it consistently excels under specific configurations, even if its overall average performance across the broader Evaluation Condition (EC) space is lowered by sub-optimal settings.
>
> This is precisely why we propose a component evaluation framework rather than a static benchmark ranking. By sampling the EC space in an unbiased manner, our goal is to characterize both a component's average performance and its sensitivity to environmental changes. Our framework naturally supports your perspective on "tunability": if a component's high variance is driven by a few interpretable EC dimensions that remain consistent across datasets, it remains highly valuable in practice.
>
> To concretely address your concern, we provide an additional table below detailing the MSE mean and standard deviation under various parameter settings of PatchTST based on Experiment 1. This table clarifies whether the observed variance stems from specific, tunable EC dimensions or chaotic interactions, further demonstrating our framework's capacity to evaluate both global robustness and practical tunability.
>
> ***Table 2**: MSE (mean±std) by learning_rate under different pred_len of PatchTST*
>
> | pred_len | lr=0.0001 | lr=0.001 |
> |----------|-----------|----------|
> | 96 | **0.2263±0.0842** | 0.2691±0.0955 |
> | 192 | 0.3026±0.0908 | **0.2945±0.1164** |
> | 336 | **0.3246±0.0982** | 0.3507±0.1101 |
> | 720 | 0.4088±0.0750 | **0.3808±0.1073** |
>
> ***Table 3**: MSE (mean±std) by d_model under different pred_len of PatchTST*
>
> | pred_len | d_model=64 | d_model=128 | d_model=256 | d_model=512 |
> |----------|------------|-------------|-------------|-------------|
> | 96 | 0.2549±0.0924 | 0.2544±0.0998 | **0.2296±0.0863** | 0.2673±0.0949 |
> | 192 | **0.2785±0.1026** | 0.2970±0.1064 | 0.2872±0.0952 | 0.3242±0.1071 |
> | 336 | **0.3142±0.0998** | 0.3524±0.0949 | 0.3377±0.1139 | 0.3484±0.1088 |
> | 720 | 0.3944±0.0753 | **0.3687±0.0947** | 0.4119±0.0866 | 0.4060±0.1135 |
>
> ### **Q3: TimeMixer Results**
> We introduce TimeMixer in `Experiment 2` as a strong baseline with multi-scale design. We has not applied our evaluation protocol to TimeMixer, so the results are incomplete. Those data from TimeMixer's best results they announced in their paper, which is not directly comparable to our evaluation results. We will clarify this point in the final version of the paper to avoid confusion.
>
> ### **W3/W4/W5: Formatting Issues**
> We thank the reviewer for pointing out these formatting issues. We will fix all formatting issues in the final manuscript to ensure that the presentation of our results is clear and consistent.
>
> ### **Mixed 1: Originality**
> We sincerely thank the reviewer for this insightful comment and for recognizing the value of our large-scale empirical efforts.
>
> To clarify the scope of our originality: the foundational theory of our statistical experiments—specifically the concepts of the Evaluation Object (EO) and Evaluation Condition (EC)—is indeed adopted from the principles of Evaluatology. However, the systematic modular decomposition of time-series forecasting architectures and the subsequent fine-grained component analysis are entirely our original contributions.

---

> > ### Author Rebuttal · Reviewer_c29R · 2026-04-03
> >
> > I thank the authors for taking the time to answer my questions. However, your answer for Q2/W2 does not fully answer what I am wondering about, since it only looks at the impact of a single variable in EC. To clarify, I'm mostly interested in whether optimizing the hyperparameters in EC are robust or brittle. By robust, I mean that the choice of optimal hyperparameters barely depends on external factors (randomness, noise, choice of dataset, ...). And by brittle, I mean situations where the choice of optimal hyperparameters are highly dependent on said external factors, making any optimization results in overfitting and being a fool's errand. I'm expecting the truth to be in middle, but I'm curious if your framework is able to extract that information.

---

> > > ### Author Response · Authors · 2026-04-04
> > >
> > > We appreciate the reviewer's further clarification. We completely agree with the reviewer’s insight that the nature of hyperparameter optimization in EC (and forecasting in general) often lies in the "middle ground". Several observations from our experiments illustrate this point:
> > >
> > > > **1. Robustness:**
> > > >
> > > > Based on additional analysis of our data, we observed that the optimal configurations for different models remain highly consistent on the Electricity dataset.
> > > >
> > > > **Specifically, for the Identity Encoder, the optimal results across all datasets are predominantly achieved with a seq_len of 512.**
> > > >
> > > > ***Table 1** Patch-wise Embedding + Transformer Encoder (PatchTST core architecture) on Electricity*
> > > > | pred_len | mse | learning_rate | d_model | seq_len | e_layers |
> > > > |-|-|-|-|-|-|
> > > > | 96 | 0.131465 | 0.001 | 512 | 512 | 3 |
> > > > | 192 | 0.146494 | 0.001 | 512 | 512 | 2 |
> > > > | 336 | 0.164374 | 0.001 | 512 | 512 | 1 |
> > > > | 720 | 0.200402 | 0.001 | 512 | 512 | 2 |
> > > >
> > > >
> > > > ***Table 2** Variate-wise Embedding + Transformer Encoder (iTransformer core architecture) on Electricity*
> > > > | pred_len | mse | learning_rate | d_model | seq_len | e_layers |
> > > > |-|-|-|-|-|-|
> > > > | 96 | 0.133547 | 0.001 | 256 | 512 | 1 |
> > > > | 192 | 0.155468 | 0.001 | 512 | 512 | 1 |
> > > > | 336 | 0.172209 | 0.001 | 256 | 512 | 1 |
> > > > | 720 | 0.197223 | 0.001 | 256 | 336 | 3 |
> > > >
> > > >
> > > > ***Table 3** Variate-wise Embedding + Identity Encoder (iLinear) on Electricity*
> > > > | pred_len | mse | learning_rate | d_model | seq_len | e_layers |
> > > > |-|-|-|-|-|-|
> > > > | 96 | 0.142832 | 0.001 | 512 | 512 | 0 |
> > > > | 192 | 0.158364 | 0.001 | 512 | 512 | 0 |
> > > > | 336 | 0.175561 | 0.001 | 512 | 512 | 0 |
> > > > | 720 | 0.213702 | 0.001 | 512 | 512 | 0 |
> > > >
> > > >
> > > > ***Table 4** Patch-wise Embedding + Identity Encoder (PatchLinear) on Electricity*
> > > > | pred_len | mse | learning_rate | d_model | seq_len | e_layers |
> > > > |-|-|-|-|-|-|
> > > > | 96 | 0.143490 | 0.001 | 512 | 512 | 0 |
> > > > | 192 | 0.157885 | 0.001 | 256 | 512 | 0 |
> > > > | 336 | 0.174540 | 0.0001 | 512 | 512 | 0 |
> > > > | 720 | 0.213784 | 0.001 | 512 | 512 | 0 |
> > > >
> > > >
> > > > **2. Brittleness:**
> > > >
> > > > We list the optimal configurations for PatchTST on ETTm1 below, where the consistency observed on the Electricity dataset is less pronounced.
> > > > Similarly, for PatchLinear on ETTm1, while a seq_len of 512 continues to perform excellently, hyperparameters such as d_model and learning_rate begin to show variations.
> > > > This demonstrates the brittleness of the optimal configurations.
> > > >
> > > > ***Table 5** Patch-wise Embedding + Transformer Encoder (PatchTST core architecture) on ETTm1*
> > > > | pred_len | mse | learning_rate | d_model | seq_len | e_layers |
> > > > |-|-|-|-|-|-|
> > > > | 96 | 0.290908 | 0.001 | 64 | 512 | 1 |
> > > > | 192 | 0.335127 | 0.0001 | 256 | 336 | 2 |
> > > > | 336 | 0.366946 | 0.001 | 128 | 192 | 2 |
> > > > | 720 | 0.414993 | 0.0001 | 64 | 512 | 1 |
> > > >
> > > > ***Table 6** Patch-wise Embedding + Identity Encoder (PatchLinear) on ETTm1*
> > > >
> > > > | pred_len | mse | learning_rate | d_model | seq_len | e_layers |
> > > > |-|-|-|-|-|-|
> > > > | 96 | 0.303390 | 0.001 | 128 | 512 | 0 |
> > > > | 192 | 0.335373 | 0.001 | 512 | 512 | 0 |
> > > > | 336 | 0.366495 | 0.0001 | 512 | 512 | 0 |
> > > > | 720 | 0.418525 | 0.0001 | 256 | 512 | 0 |
> > > >
> > >
> > > In summary, our framework is inherently capable of extracting this information. However, as shown in the examples above, we observe both Robustness and Brittleness within our EC space. Analyzing every dataset and model combination would yield many specific findings, but such conclusions are often difficult to transfer to other datasets or models to provide general guidance for the time-series forecasting field.
> > >
> > > Consistent with our response to Reviewer `kGaa` (Q2), we believe that evaluating the stability of a component is more valuable than searching for dataset-specific optimal hyperparameters. We hope these examples answer your questions. We are happy to provide more detailed analyses for other datasets and models if the reviewer is interested.

---

### Official Review · Reviewer_kGaa · 2026-03-11

**Soundness:** 3
**Presentation:** 3
**Significance:** 3
**Originality:** 3
**Overall Recommendation:** 5
**Confidence:** 4

**Summary:**

The paper introduces CombinationTS, a framework that tries to understand existing state-of-the-art time-series forecasting models from a modular perspective. Instead of proposing another forecasting architecture, the authors break the forecasting pipeline into several components, such as input transformation, embedding, encoder, decoder, and output transformation, and study them separately. The framework evaluates these components under a shared evaluation condition space and reports statistics like marginalized effectiveness and stability rather than relying on single-run benchmark results.

Using this setup, the paper runs experiments across several forecasting datasets and many combinations of modules. One of the main observations is that representation-related design choices, such as embedding or data view, appear to have a larger impact than the encoder architecture itself. In particular, the results suggest that when the representation is well designed, even a simple identity encoder can achieve performance comparable to more complex backbones.

**Compliance With Llm Reviewing Policy:**

Affirmed.

**Final Justification:**

My main concerns have been addressed. In particular, the discussion on hyperparameter sensitivity significantly resolves my confusion. The clarification on the “Identity” encoder and the paired EC sampling protocol also improves clarity. I am more confident in the work and am happy to increase my score.

**Key Questions For Authors:**

1. Why is the term “identity” used to denote the case without an encoder?

2. How sensitive are models such as PatchTST, DLinear, and SimpleTM to hyperparameters? In particular, how does their performance vary across the parameter space? I would also be interested in seeing further analysis on how different module combinations affect hyperparameter sensitivity.

3. Some tables in the appendix appear without being referenced in the main text.

**Limitations:**

Yes

**Strengths And Weaknesses:**

Strengths

1. The paper is well motivated. It raises a clear concern about how performance improvements in time-series forecasting models are currently evaluated. In particular, the paper argues that existing benchmarking practices often mix together multiple factors such as representation design, encoder architecture, and hyperparameter choices, making it difficult to attribute where the actual performance gains come from. Framing this as a modular attribution problem is reasonable and clearly explained.

2. Interesting empirical findings supported by extensive experiments. The empirical results are quite interesting and appear to be supported by a relatively large set of experiments across datasets and module combinations. Some of the observations are particularly thought-provoking. For example, the results suggest that on datasets such as ETTh1, complex encoders may not be necessary once the representation is properly designed, and simple identity mappings can perform competitively.

3. Promising direction of applying evaluatology-style analysis. The idea of applying the evaluatology perspective to time-series forecasting evaluation is promising. Instead of relying on fixed “fair settings” or searching for the best hyperparameters, the paper analyzes performance over a space of evaluation conditions and reports distributional statistics such as marginalized effectiveness and stability.

4. Open source code will be helpful for community.

Concerns
1. The description of the sampling protocol and how paired EC sampling is implemented could be clearer.

2. Table 6 is significant enough that it belongs in the main body instead of the appendix.

3. The experimental setup section could be better organized. The current structure feels somewhat confusing and may benefit from clearer organization.

---

> ### Author Rebuttal · Authors · 2026-03-31
>
> We sincerely thank the reviewer for the thoughtful feedback and for recognizing the significance of our modular attribution framework and the compelling nature of our empirical findings. Below, we provide detailed responses to your specific questions and concerns.
>
> ### **Q1: Identity**
>
> We thank the reviewer for this question. Indeed, the "Identity" encoder simply represents the absence of an encoder and contains no learnable parameters. We chose the term "Identity" primarily to maintain a consistent structural comparison with other instantiated encoders within our modular framework.
>
> Furthermore, this naming convention aligns with established practices in the deep learning community. For instance, PyTorch utilizes the *torch.nn.Identity* module as a completely transparent placeholder that outputs its input without modification. Therefore, we adopted this terminology to ensure structural elegance in our framework and to remain consistent with standard programming conventions in the AI community.
>
> ### **Q2: Hyperparameter Sensitivity**
> We sincerely thank the reviewer for this insightful question regarding hyperparameter sensitivity and the behavior of models like PatchTST, DLinear, and SimpleTM across the parameter space.
>
> To concretely illustrate how performance varies across the parameter space, we provide an analysis of the **Patch Embedding + Transformer Encoder + Linear Decoder** combination (the core architecture of PatchTST) as a representative example. The tables below detail the MSE mean and standard deviation under various hyperparameter settings across six datasets.
>
> **Patch Embedding + Transformer Encoder MSE Analysis**
>
> *Table 1: MSE (mean±std) by learning_rate under different pred_len*
>
> | pred_len | lr=0.0001 | lr=0.001 |
> |:---|---:|---:|
> | 96 | **0.2263±0.0842** | 0.2691±0.0955 |
> | 192 | 0.3026±0.0908 | **0.2945±0.1164** |
> | 336 | **0.3246±0.0982** | 0.3507±0.1101 |
> | 720 | 0.4088±0.0750 | **0.3808±0.1073** |
>
> *Table 2: MSE (mean±std) by d_model under different pred_len*
>
> | pred_len | d_model=64 | d_model=128 | d_model=256 | d_model=512 |
> |:---|---:|---:|---:|---:|
> | 96 | 0.2549±0.0924 | 0.2544±0.0998 | **0.2296±0.0863** | 0.2673±0.0949 |
> | 192 | **0.2785±0.1026** | 0.2970±0.1064 | 0.2872±0.0952 | 0.3242±0.1071 |
> | 336 | **0.3142±0.0998** | 0.3524±0.0949 | 0.3377±0.1139 | 0.3484±0.1088 |
> | 720 | 0.3944±0.0753 | **0.3687±0.0947** | 0.4119±0.0866 | 0.4060±0.1135 |
>
> As observed, the hyperparameter sensitivity is highly dynamic. However, based on our extensive experimental data across the Evaluation Condition (EC) space, we did not observe any consistent, universally applicable conclusions regarding optimal hyperparameter selection.
> Furthermore, we found that hyperparameter tuning heuristics derived from specific datasets are notoriously difficult to transfer to other datasets due to varying temporal dynamics and data characteristics. Consequently, conducting an exhaustive, dataset-specific analysis of hyperparameter combinations for every model (such as PatchTST, DLinear, and SimpleTM) incurs a prohibitively high computational overhead. Because these dataset-specific insights fail to generalize, we believe the limited conclusions drawn do not justify the massive experimental cost.
>
> This limitation is precisely why our framework focuses on *distributional robustness* rather than pinpointing dataset-specific optimal hyperparameters. Instead of exhaustively tuning individual models, our approach efficiently characterizes a module's overall sensitivity (variance) across the parameter space, proving that evaluating a component's stability is more reliable and valuable than chasing non-transferable hyperparameter peaks.
>
> ### **C1: Clarification of Sampling Protocol**
>
> To clarify, the core principle of our "paired EC sampling" is that all Evaluated Objects (EOs) are assessed under the **exact same subspace** of Evaluation Conditions (ECs).
>
> As outlined in `Sections 3.3` and `Section 4.1.1`, the implementation of this protocol proceeds as follows:
> 1. **Single Stratified Sampling:** We perform a single, stratified Monte Carlo sampling over the entire theoretical EC space to generate a fixed subspace of specific configurations.
> 2. **Strict Pairing:** Every modular variant (EO) is subsequently evaluated against this identical set.
> 3. **Handling Inactive Dimensions:** To accommodate structural differences between EOs (e.g., the parameter-free Identity encoder does not utilize the *encoder_layers* hyperparameter), we employ a *Common Random Numbers* design. If a sampled EC dimension is inactive for a specific EO, it is simply ignored. Crucially, the rest of the configuration (e.g., learning rate, data view) remains perfectly aligned, preserving the sampled EC's identity.
>
> ### **Q3/C2/C3: Formatting Issues**
> We thank the reviewer for pointing out these formatting issues. We will fix all formatting issues in the final manuscript to ensure that the presentation of our results is clear and consistent.

---

> > ### Author Rebuttal · Reviewer_kGaa · 2026-04-02
> >
> > Thank you for the clear and detailed rebuttal. My main concerns have been addressed. In particular, the discussion on hyperparameter sensitivity significantly resolves my confusion. The clarification on the “Identity” encoder and the paired EC sampling protocol also improves clarity. I am more confident in the work and am happy to increase my score.

---

> > > ### Author Response · Authors · 2026-04-02
> > >
> > > We appreciate your insightful feedback and the time you put into our manuscript. We are happy to hear that our responses cleared up your concerns. We will update the manuscript according to your advice. Thanks again for your constructive support.

---

### Official Review · Reviewer_gxzB · 2026-03-16

**Soundness:** 3
**Presentation:** 3
**Significance:** 4
**Originality:** 3
**Overall Recommendation:** 5
**Confidence:** 3

**Summary:**

The paper proposes CombinationTS, a modular evaluation framework for time-series forecasting (TSF). The authors identify two core methodological deficits: the Attribution Gap (monolithic model evaluation conflates data view and encoder contributions) and the Benchmarking Crisis (point-estimate SOTA claims are statistically fragile). CombinationTS decomposes the TSF pipeline into five orthogonal stages and evaluates each component under a shared probabilistic Evaluation Condition (EC) space via stratified Monte Carlo sampling with a Common Random Numbers design, reporting marginalized effectiveness μ and stability σ². Three empirical findings emerge across ~100 architectural variants on six benchmarks: the Identity Paradox (parameter-free Identity encoder matches or outperforms complex Transformers when embedding is well-designed); Conditional Utility of Explicit Priors (cyclic priors help consistently; naive decomposition without cross-component interaction often hurts); and Spectral Stability (frequency-domain encoders primarily reduce variance rather than improving mean effectiveness). Code and full experimental logs are released.

**Compliance With Llm Reviewing Policy:**

Affirmed.

**Key Questions For Authors:**

1. Seed robustness: Could the authors provide even a limited multi-seed analysis (e.g., 3–5 seeds on one dataset) to verify that the Identity Paradox is not seed-sensitive? Without this, σ² risks being misread as a broader robustness measure than it is.

2. Decoder constraint: Does Identity's advantage over Transformer persist when a more expressive decoder is permitted? Even one dataset-level verification would significantly qualify or reinforce the Identity Paradox claim.

3. Spectral generalizability: Are the Experiment 3 conclusions specific to the SimpleTM-vs-iTransformer pair, or do they hold across other spectral encoder designs under the same EC protocol?

**Limitations:**

CombinationTS makes a genuine and timely methodological contribution. My main reservations are presentational: the exclusion of seed variance from σ² should be explicitly scoped as a limitation rather than left implicit, and the Identity Paradox should be framed as benchmark-conditional rather than universal—calling deep encoders "over-parameterized noise generators" is an overreach given the narrow benchmark scope. These are framing issues; the underlying framework and findings remain sound.

**Strengths And Weaknesses:**

Strengths:
1. Well-motivated problem framing. The attribution gap and benchmarking crisis are real, underappreciated problems. The paper provides a clean conceptual unification of both under a single probabilistic evaluation paradigm.
2. Principled evaluation protocol. Paired EC sampling (CRN design) with distributional metrics (μ, σ²) is a meaningful upgrade over fixed-config point estimates. Stratified sampling prevents domain imbalance. Full CIs are reported.
3. Compelling empirical findings. The Identity Paradox—Transformers failing to outperform a parameter-free pass-through across all six datasets under paired conditions—is striking and independently corroborated across three experiments. The finding that naive decomposition without interaction degrades performance is equally actionable.
4. Reproducible and open-source. Releasing the modular library and full experimental logs meaningfully supports the paper's own reproducibility standard.

Weaknesses:
1. Internal inconsistency on seed robustness. Section 3.2 lists random seeds as part of the EC definition, yet the protocol fixes the seed throughout and explicitly disclaims seed robustness evaluation. The resulting σ² measures only hyperparameter sensitivity, not the broader training stochasticity the framing implies. This is a notable gap given the paper's core motivation.
2. Benchmark scope limits generalizability. All experiments use six standard LTSF benchmarks dominated by periodic, low-noise signals. Whether the Identity Paradox and the Embedding > Encoder hierarchy hold on more complex or non-stationary domains is untested and unaddressed.
3. Decoder constraint introduces attribution bias. Fixing the decoder to Shared Linear in Experiment 1 may systematically disadvantage Transformers that benefit from more expressive decoders, potentially inflating the Identity Paradox effect.
4. Spectral conclusion rests on a single comparison. Experiment 3's spectral stability claim derives from one architectural pair (iTransformer vs. SimpleTM). A broader set of spectral-vs-temporal pairs would substantially strengthen the conclusion.

---

> ### Author Rebuttal · Authors · 2026-03-31
>
> We sincerely thank the reviewer for the highly constructive feedback and for recognizing the value of our conceptual framing, our principled probabilistic protocol, and the compelling nature of our empirical findings. We also deeply appreciate your insightful suggestions regarding the boundaries of our claims and our experimental scope. Below, we address your specific questions in detail.
>
> ### **Q1/W1: Seed robustness**
>
> We appreciate the reviewer's attention to the seed robustness aspect of our evaluation. While multi-seed analysis is crucial for traditional point estimation, our distributional evaluation framework relies on extensive sampling within the Evaluation Condition (EC) space to capture the average performance and variability of the model. The variance introduced by random seeds is effectively marginalized and smoothed out through this comprehensive sampling process. We conducted an additional analysis to compare the MSE statistics across three different random seeds (333, 2025 and 2026) for the PatchLinear and PatchTST model on ETTh1 dataset. The results are shown in the tables below:
>
> ***Table 1**: PatchLinear MSE statistics across different random seeds on ETTh1 dataset.*
>
> | Metric | seed 333 | seed 2025 | seed 2026 |
> |-|-|-|-|
> | $\mu$ | 0.425837 | 0.425889 | 0.425468 |
> | $\sigma$ | 0.037791 | 0.037417 | 0.037166 |
>
> ***Table 2**: PatchTST MSE statistics across different random seeds on ETTh1 dataset.*
>
> | Metric | seed 333 | seed 2025 | seed 2026 |
> |-|-|-|-|
> | $\mu$ | 0.448368 | 0.448524 | 0.445542 |
> | $\sigma$ | 0.057974 | 0.056290 | 0.050897 |
>
> As shown above, the results remain highly consistent across different seeds. This confirms that our evaluation is robust to seed-induced stochasticity, providing a stable foundation for reliable and reproducible performance.
>
> ### **Q2: Decoder constraint**
> We acknowledge the reviewer's insightful question regarding the decoder constraint.
> According to `Chen et al. [1]`, the Variate-independent Decoder we adopted in our study has been shown to be effective in Transformer-based time series forecasting models.
> Their analysis indicates that introducing a more expressive decoder, such as adding convolutional layers to capture inter-variable interactions, does not significantly improve performance.
>
> To simplify our experiments and focus on the core contributions of our work, we chose widely used decoders for our evaluation. For variate-wise embedding, the linear decoder we adopt is the decoder of iTransformer, which actually is the same as the one used by PatchTST.
>
> [1] Chen Y, Céspedes N, Barnaghi P. A closer look at transformers for time series forecasting: Understanding why they work and where they struggle[C]//Forty-second International Conference on Machine Learning. 2025.
>
> ### **Q3: Spectral generalizability**
>
> We appreciate the reviewer's interest in the spectral generalizability of our findings.
> Our work focused on the modular framework for understanding time-series forecasting models.
> So we choose the SimpleTM and iTransformer as representative examples of spectral encoder designs to demonstrate the effectiveness of our evaluation protocol.
> The evaluation protocol can be applied to other spectral methods, and we encourage future research to explore this aspect further.
>
> ### **W1: Inconsistency on Seed Robustness**
> We appreciate the reviewer's careful reading. We would like to clarify that `Section 3.2` presents our module-level evaluation framework and defines the *complete* Evaluation Condition (EC) space, which theoretically includes random seeds.
>
> In the experimental section, however, we intentionally simplified the practical EC space. As we demonstrated with empirical evidence in our response to `Q1`, the choice of random seed does not affect our experimental results. Therefore, we fixed the seed to a specific value to focus our computational resources.
>
> ### **W2: Dataset Diversity**
>
> We appreciate the reviewer's constructive feedback on dataset diversity. In our study, we evaluated our framework on six standard benchmarks that are widely used in the time-series forecasting community.
>
> We agree that understanding how these findings transfer to more complex domains is important.
> To address this, we conducted additional experiments on the **Exchange-Rate** dataset, which is complex and non-stationary. We performed a targeted hyperparameter search for two of our framework instantiations: **PatchLinear** (Patch-wise embedding + Identity encoder) and **iLinear** (Variate-wise embedding + Identity encoder).
>
> ***Table 3**: MSE results on Exchange-Rate dataset for PatchLinear and iLinear across different prediction lengths.*
> | pred_len | iLinear | PatchLinear | iTransformer | PatchTST |
> |-|-|-|-|-|
> | 96 | 0.094 | **0.083** | 0.086 | 0.088 |
> | 192 | 0.187 | **0.176** | 0.177 | **0.176** |
> | 336 | 0.343 | 0.325 | 0.331 | **0.301** |
> | 720 | 0.874 | **0.844** | 0.847 | 0.901 |
> | **Avg** | 0.375 | **0.357** | 0.360 | 0.367 |

---

> > ### Author Rebuttal · Reviewer_gxzB · 2026-04-04
> >
> > Thank you for the multi-seed results and the additional Exchange-Rate experiments — these do a good job addressing my concerns on seed robustness and generalizability. The explanation on the decoder constraint is also reasonable, and I'm happy to keep my score as is.

---

> > > ### Author Response · Authors · 2026-04-04
> > >
> > > We sincerely thank the reviewer for the positive feedback and for acknowledging our efforts in the rebuttal. We are glad that the additional multi-seed results and Exchange-Rate experiments successfully addressed your concerns regarding robustness and generalizability.
> > >
> > > Your insightful comments on the decoder constraints and experimental validation have significantly helped us improve the rigor of our work. We appreciate your support for our paper.

---

### Decision · Program_Chairs · 2026-04-30

**Decision:**

Accept (regular)

**Comment:**

The paper introduces CombinationTS, a modular evaluation framework for time-series forecasting that decomposes and evaluates  various components of  state-of-the-art time-series forecasting architectures such as input transformation, embedding, encoder, decoder, and output transformation, and study them separately. The framework evaluates these components under a shared evaluation condition space and reports statistics like marginalized effectiveness and stability rather than relying on single-run benchmark results. Most reviewers agreed that the paper is well motivated and timely, and  raises important questions  about how the community should interpret architectural claims and performance improvements for time-series forecasting models. Some reviewers did raise concerns on the fact that this paper uses  a subset of the standard long horizon forecasting datasets for evaluation, which are small and prone to overfitting issues (as opposed to  more modern and comprehensive evaluation datasets like Gift-Eval). Nevertheless, overall the paper makes a timely and important methodological contribution to the time series community, and the AC is happy to recommend acceptance of the paper.